# TensorSLM: Sub-billion Parameter Language Model Compression for Low-end Devices based on Tensor-Train Decomposition

## Abstract

The Small Language Models (SLMs, or on-device LMs) (Lu et al., 2024) is a concept corresponding to the Large Language Model (LLM), which has significantly fewer parameters and is typically deployed on low-end devices, like mobile phones (Liu et al., 2024) and single-board computers (e.g. Raspberry Pi). Unlike LLMs, which utilize the increasing model size for better generalization, SLMs are expected to adjust the exact deployment environment changes. Furthermore, most edge applications have battery life concerns, which have never been considered in the GPU servers for data centres. Targeting these two issues, this paper focuses on the *token embedding compression* for **adaptivity** and **low energy** requirements in edge applications. We propose a **training-free** model compression approach based on the Tensor-Train Decomposition (TTD), whereby each pre-trained token embedding vector is converted into a lower-dimensional Matrix Product State (MPS). We then comprehensively investigate the low-rank structures extracted by this approach, regarding the compression ratio, language task performance, latency and energy consumption on a typical low-end device (i.e. Raspberry Pi). Taking the sub-billion parameter versions of GPT-2/Cerebres-GPT and OPT as examples, the model compressed with our approach can achieve a comparable language task performance to the original model with around $2.0\times$ embedding layer compression, while the energy consumption of single query drops by half.

## 1 Introduction

Modelling complex language patterns and solving complex language tasks are two of the primary reasons that LLMs have attracted considerable attention these years. While the large-scale language model track thrives on having larger sizes and solving more difficult tasks, another track is considering putting such capable models on lower-end devices. These models are called small language models (SLMs) (Lu et al., 2024) or on-device language models (Liu et al., 2024; Mehta et al., 2024; hfs).

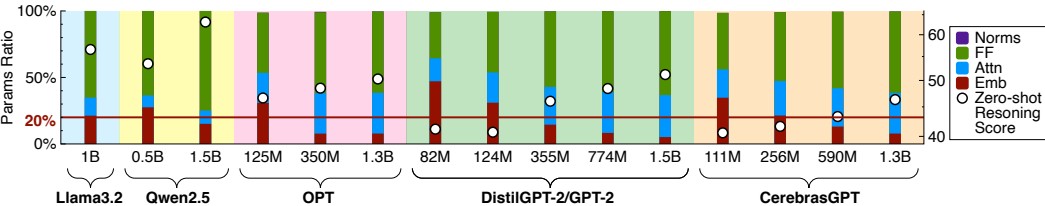

Figure 1: The parameter ratio of Norms (including layer norms), feed-forward layers (FF), attention layers (Attn), and embedding layers (Emb), and the average zero-shot reasoning score (on the tests of HellaSwag (Zellers et al., 2019), ARC-easy/-challenge (Clark et al., 2018), BoolQ (Clark et al., 2019) and PIQA (Bisk et al., 2020)) of several open-source model series. Inside a model series, smaller models have a higher token embedding layer ratio and lower feed-forward layer ratio. The attention layer ratio roughly maintains the still with model size changing (within the same model series).

SLMs may have less than one billion parameters (Mehta et al., 2024; Liu et al., 2024; Laskaridis et al., 2024). Though such a size is already a few tenths or even hundreds of what common LLMs usually are, for some low-end devices, it can still be a burden. As listed in (Liu et al., 2024, Fig. 2), some prevalent mobile devices (e.g. iPhone 14 and iPhone 15) only have 6GB DRAM. For some small language models like Gemma2-2B, it causes a system crash if running its uncompressed version on Raspberry Pi-5 with 8GB DRAM.

Compared with LLMs, SLMs on low-end devices have different layer compositions of the model and different on-board operations due to the absence of server-level GPUs. As shown in Fig. 1, around half of investigated open-source models have more than 20% parameters attributed to the token embedding layers, which is consistent with the statements in (Liu et al., 2024, 2.2.3). Also, since no server-level GPU is on board to support massive parallel operations for matrix multiplication, block-wise approaches that rely on parallelism (Dao et al., 2022; Qiu et al., 2024) are not suitable.

To this end, this paper proposes TensorSLM, a tensor-based approach to compress SLMs for low-end devices (i.e. Raspberry Pi without GPU). Together with matrix-based low-rank approaches (Chen et al., 2018a; Hrinchuk et al., 2020; Lioutas et al., 2020; Acharya et al., 2019; Chen et al., 2021; Hsu et al., 2022; Dao et al., 2022; Qiu et al., 2024), this kind of approach forms a broader field named low-rank factorization. The comparison of these works regarding methodologies (e.g. matrix/tensor, with/without training) and applications (e.g. high-end/low-end devices, large/small models) are clarified in Sec. 3. Compared with two-dimensional matrices or their finer-grained block-wise forms (Chen et al., 2018a; Dao et al., 2022), higher-order tensors provide more diverse representation alternatives with their inter-order information, which is more suitable for small-size models to solve complex tasks. This superiority is more pronounced if there is no fine-tuning data to adjust the model parameters for the exact application environments.

The contributions of this paper are summarised as follows:

1. We provide a systematic analysis of LLMs on high-end GPU servers and SLMs on low-end edge devices to address the two unique requirements of SLM compression: *adaptability* and *low energy*.

2. As far as we know, we are the first to compress SLMs for low-end device use cases, with low-rank factorization. We adjust Tensor-Train Decomposition for non-parallel operations in the forwarding passes, where block-wise approaches (Dao et al., 2022; Qiu et al., 2024) are incompetent.

3. We give the measured latency and estimated energy consumption of SLMs on the typical low-end device, Raspberry Pi 5[1], and it comes out that our approach reduces half of the inference energy with negligible latency increase.

4. We evaluated both simple and complex language tasks. We found that our tensor-based approach is better at unprompted and unconstrained question answering than the matrix-based SVD approach, and herein shed light on selecting appropriate algebraic structures according to the tasks.

## 2 UNIQUE REQUIREMENTS OF SLM APPLICATIONS

### 2.1 ADAPTABILITY

Unlike the current LLM applications, which are mostly running on high-end GPU servers (e.g. in the data centres with numerous NVIDIA A100), SLMs are mainly for edge (or mobile) applications that require adapting to the environment with limited resources on lower-end devices. A common approach to adapting to the dynamic environment is updating the vocabulary according to the changes in input text distribution Chen et al. (2018a). The reasons for this distribution change vary from case to case. For example, new user registration, or the frequently used tokens update with the users' changing daily lives.

To cope with the ever-changing input tokens and vocabulary, a straightforward strategy is to build a could-edge system, as shown in 2, which is similar to the workflows in the field of edge computing, e.g. (Laskaridis et al., 2024, Fig.1). There are two kinds of devices in this workflow: 1) the central server, which is possibly a server in public or private cloud services, or a higher-end personal computer, and 2) the low-end edge device. In this paper, we only talk about a typical edge device -

---

[1] https://www.raspberrypi.com/products/raspberry-pi-5/

Figure 2: The workflow of SLM compression in edge computing scenario with TensorSLM.

Raspberry Pi. Over a fairly long period (e.g. months or years), the central server only communicates with the edge device once to provide a brand-new pre-trained language model. Afterwards, the edge device should update the vocabulary on board according to the changes in the environment.

A detailed explanation of Fig. 2 is as follows:

**Step 1.** The central server compresses the whole token embedding matrices on the token embedding level, according to Alg. 1.

**Step 2.** The compressed vocabulary and other parts of the language model (e.g. the decoder) are downloaded and then deployed on a low-end device.

**Step 3.** During the application runs, the vocabulary updates for two cases:

    (a) a new token is required according to the actual application requirements, it will be registered by the service on the edge device. Jump to **Step 4**.

    (b) an old token is required to be removed (e.g. it has not been used for a long time), the edge device simply deletes the corresponding token embedding vector. Meanwhile, the application deregisters this token.

**Step 4.** The low-end device compresses the added token embedding vector as described in Alg. 1.

**Step 5.** The current vocabulary of the language model. The compression process of a single token embedding follows a pipeline of ① tensorization and ② decomposition.

## 2.2 LOW ENERGY CONSUMPTION

From the workload of the high-end GPU servers (e.g. those equipped with NVIDIA A100) and low-end edge devices (e.g. Raspberry Pi 5) described in Sec. 2.1, we know that the edge device only takes charge of light-weight essential tasks, since it has strict limitations in computation, memory and communication. Furthermore, since battery life directly impacts the user experience, energy consumption is also a significant concern.

The actual energy consumption of a device depends on various factors, like the semiconductor temperature, system workload, operating environment, etc. Thus, it is hard to precisely calculate the

Table 1: Approximate energy consumption of different operations (1nJ=1000pJ). For servers, communication with the wired network (e.g. ethernet or optical fibre) is preferred; for edge devices, it is preferred to use wireless networks (e.g. Wi-Fi or cellular network).

| Device Type | Computation (pJ/`float32`) | | Memory (pJ/`float32`) | Communication (nJ/`float32`) | |
|---|---|---|---|---|---|
| | **Add** | **Mult** | | **wired** | **wireless** |
| Raspberry Pi 5 (Cortex-A76 CPU) | 1.0-2.5 | 1.2-3 | 70-260 | 50-350 | 400-6000 |
| GPU server (A100 GPU) | 5-12 | 6-15 | 100-450 | | |

Table 2: Comparison with TensorSLMs and relevant research.

| Study on LM compression or relevant low-rank factorization | Device | | Training ? | Algebra Structure | | Layer | | Focused Size | |
|---|---|---|---|---|---|---|---|---|---|
| | high-end | low-end | | matrix | tensor | Emb | Linear | large | small |
| Chen et al. (2018a) | ✓ | | ✓ | ✓ | | ✓ | | | ✓ |
| Hrinchuk et al. (2020) | ✓ | | | | ✓ | ✓ | | | ✓ |
| Wang et al. (2023) | ✓ | | ✓ | ✓ | | ✓ | | | ✓ |
| Bałazy et al. (2021) | | ✓ | ✓ | ✓ | | ✓ | | | ✓ |
| Liu et al. (2015) | - | | ✓ | ✓ | | | | - | |
| Chen et al. (2018b) | ✓ | | ✓ | ✓ | | ✓ | | | ✓ |
| Yuan et al. (2023) | ✓ | | | | ✓ | | ✓ | ✓ | |
| Hsu et al. (2022) | ✓ | | ✓ | ✓ | | | ✓ | | ✓ |
| Chekalina et al. (2023a) | ✓ | | ✓ | | ✓ | | ✓ | ✓ | |
| Lin et al. (2024) | ✓ | | | | | | ✓ | ✓ | |
| Dao et al. (2022) | ✓ | | ✓ | ✓ | | | ✓ | | |
| Qiu et al. (2024) | ✓ | | ✓ | ✓ | ✓ | | ✓ | | |
| Liu et al. (2024) | ✓ | | ✓ | - | | | | | ✓ |
| TensorSLM (ours) | | ✓ | | | ✓ | ✓ | | | ✓ |

exact energy consumption of an algorithm on a certain hardware. However, we can still estimate the range of energy consumption in the system as Tab. 1, where we can have the following remarks:

**Remark 1.** Memory operations are more "expensive" than computation in terms of energy.

**Remark 2.** Non-essential communication should be avoided for energy concerns.

Our workflow Fig. 2 has already satisfied Rem. 2. For Rem. 1, if real-time is *not* the most important concern in the edge application, it is reasonable to "exchange" memory with computation for longer battery life. We will later discuss and evaluate this point in Sec. 5.1 and Sec. 6.

## 3 WHY NOT EXISTING SOLUTIONS?

The field of language model compression with low-rank factorization has been booming in recent years. The recent relevant works are summarized inTab. 2. We can observe that for the current existing works, some are specialized for embedding layers (Chen et al., 2018a; Hrinchuk et al., 2020; Wang et al., 2023; Bałazy et al., 2021; Acharya et al., 2019; Liu et al., 2015) while others are not (Chekalina et al., 2023b; Chen et al., 2021; Hsu et al., 2022; Dao et al., 2022; Qiu et al., 2024). However, all of these require an extra training process, such as fine-tuning, meta-learning (Chen et al., 2018a; 2021; Hsu et al., 2022; Bałazy et al., 2021; Liu et al., 2015; Dao et al., 2022; Wang et al., 2023; Qiu et al., 2024) and training from scratch (Hrinchuk et al., 2020; Chekalina et al., 2023b).

There are two limitations to this extra training: 1) extra training involves additional computation and training data, which may be unavailable for low-end devices; 2) training the language model from scratch discards the valuable knowledge stored in the weights of the original models. However, we only focus on training-free low-end device applications. For a more detailed discussion of these relevant works, please refer to Appx. B.

## 4 PRELIMINARIES

This section gives the essential concepts related to tensor, tensor operations and Tensor-Train Decomposition. A more complete introduction about tensors can be found in Appx. A.

**Order-N Tensor.** An order-$N$ real-valued tensor, $\mathcal{A}$, is a high-dimensional matrix (or multi-way array), denoted by $\mathcal{A} \in \mathbb{R}^{I_1 \times \cdots \times I_N}$, where $N$ is the order of the tensor (i.e., number of its modes), and $I_k$ ($1 \leq k \leq N$) is the size (i.e., the dimension) of its $k$-th mode. In this sense, matrices (denoted

---

**Algorithm 1:** `TT_SVD`(Oseledets, 2011) for a Single Token Embedding Compression

---

**Input**  : 1. $d$-dimensional token embedding vector $\mathbf{x} \in \mathbb{R}^d$, approximation accuracy $\epsilon$;
             2. Tensor dimension $\{I_1, I_2, \ldots, I_N\}$ and TT ranks $\{r_0, r_1, \ldots, r_N\}$
**Output**   : TT cores $\mathcal{G}^{(1)}, \ldots, \mathcal{G}^{(N)}$
**Initialize** : Tensor $\mathcal{X} \leftarrow$ `reshape`$(\mathbf{x}, [I_1, I_2, \ldots, I_N])$,
             temporary matrix $\mathbf{Z} \leftarrow$ `reshape`$(\mathcal{X}, [r_0 I_1, \prod_{j=2}^{N} I_j])$,
             truncation parameter $\delta = \frac{\epsilon}{\sqrt{N-1}} \|\mathcal{X}\|_F$

1 **for** $k = 1$ *to* $N - 1$ **do**
2 |   $\mathbf{U}, \mathbf{S}, \mathbf{V}, \mathbf{E} \leftarrow$ `truncSVD`$(\mathbf{Z}, \delta, r_k)$        `// s.t.` $\mathbf{U} \in \mathbb{R}^{r_{k-1} I_k \times r_k}$, $\|\mathbf{E}\|_F \leq \delta$
3 |   $\mathcal{G}^{(k)} \leftarrow$ `reshape` $(\mathbf{U}, [r_{k-1}, I_k, r_k])$        `// get` $k$`th TT core`
4 |   $\mathbf{Z} \leftarrow$ `reshape` $\left(\mathbf{S}\mathbf{V}^T, [r_k I_{k+1}, \prod_{j=k+2}^{N} I_j])\right)$        `//` $\mathbf{S}\mathbf{V}^T \in \mathbb{R}^{\prod_{i=k+2}^{N} I_i}$
5 $\mathcal{G}^{(N)} \leftarrow \mathbf{Z}$
6 **return** $\mathcal{G}^{(1)}, \mathcal{G}^{(2)}, \ldots, \mathcal{G}^{(N)}$

---

as $\mathbf{A} \in \mathbb{R}^{I_1 \times I_2}$) can be seen as order-2 tensors ($N = 2$), vectors (denoted as $\mathbf{a} \in \mathbb{R}^I$) can be seen as order-1 tensors ($N = 1$), and scalars (denoted as $a \in \mathbb{R}$) are order-0 tensors ($N = 0$).

**Tensor-Train Decomposition (TTD).**   The most common Tensor-Train Decomposition (Oseledets, 2011) formats a tensor into a Matrix Product State (MPS or TT-MPS) form, which applies the Tensor-Train Singular Value Decomposition (TT-SVD) algorithm (described in Appx. A.3) to an order-$N$ tensor, $\mathcal{X} \in \mathbb{R}^{I_1 \times I_2 \times \cdots \times I_N}$. This results in $N$ smaller 2-nd or 3-rd order tensors, $\mathcal{G}^{(k)} \in \mathbb{R}^{r_{k-1} \times I_k \times r_k}$ for $k = 1, \ldots, N$, such that

$$\mathcal{X} \approx \mathcal{G}^{(1)} \times_2^1 \mathcal{G}^{(2)} \times_3^1 \mathcal{G}^{(3)} \times_3^1 \cdots \times_3^1 \mathcal{G}^{(N)}. \tag{1}$$

Tensor $\mathcal{G}^{(1)}, \ldots, \mathcal{G}^{(N)}$ are referred to as the tensor cores, while the set $\{r_0, r_1, \ldots, r_N\}$ represents the TT-rank of the TT decomposition ($r_0 = r_N = 1$).

## 5 METHODOLOGY

This section clarifies the technical cornerstones of our approach. A practical pipeline of our approach is depicted in Fig. 2. The whole vocabulary is processed on higher-end servers, while inference and vocabulary update happens on lower-end edge devices.

### 5.1 INDIVIDUAL EMBEDDING VECTOR COMPRESSION

For the compression of the embedding matrix, rather than decomposing the whole embedding weight matrix, we propose to decompose each embedding vector. The lower half of Fig. 2 is a simplified illustration of such a process, with a detailed description in Alg. 1.

**Tensorization.**   Each token embedding $\mathbf{x} \in \mathbb{R}^d$ is reshaped (or folded, tensorized, as in Appx. A.3) into an order-$N$ tensor. Denote `reshape`$(\cdot)$ as the reshape function, $\mathcal{X} =$ `reshape`$(\mathbf{x}, \{I_1, I_2, \ldots, I_N\})$ and $\mathcal{X} \in \mathbb{R}^{I_1 \times \cdots \times I_N}$ such that $d = \prod_{k=1}^{N} I_k$. In the example in Fig. 2, the token embedding vector $\mathbf{x}$ is a 27-dimensional vector, $d = 27$. In this way, vector $\mathbf{x}$ is reshaped into an order-3 ($N = 3$) tensor $\mathcal{X}$, with tensor size for each mode $I_1 = I_2 = I_3 = 3$.

**Tensor Decomposition.**   Tensor $\mathcal{X}$ is then decomposed and stored in a Matrix Product State (MPS) form as $\mathcal{X} \approx \mathcal{G}^{(1)} \times_3^1 \cdots \times_3^1 \mathcal{G}^{(N)}$, with hyperparameters as TT ranks $r_0, r_1, \ldots, r_N$. For the case in Fig. 2, the MPS cores are $\mathcal{G}^{(1)}, \mathcal{G}^{(2)}, \mathcal{G}^{(3)}$, with TT ranks $r_0 = r_1 = r_2 = r_3 = 1$. In other words, instead of storing the entire token embedding vector $\mathbf{x} \in \mathbb{R}^d$, we store the corresponding MPS cores, $\mathcal{G}^{(k)} \in \mathbb{R}^{r_{k-1} \times I_k \times r_k}$, for $k = 1, \ldots, N$. The parameter count of the MPS cores $\{\mathcal{G}^{(k)}\}$ is $\sum_{k=1}^{N} |\mathcal{G}^{(k)}| = \sum_{k=1}^{N} r_{k-1} I_k r_k$, where $|\cdot|$ represents the parameter count.

A more detailed explanation of individual token embedding compression is given in Alg. 1, and its cornerstone `TT_SVD` is further described in Alg. 2 (in Appx. A.3), where $\|\cdot\|_F$ denotes the

Table 3: Computation and memory complexity during the compression (Sec. 5.1) and inference(Sec. 5.2) of TensorSLM. $\mathcal{M}_{\texttt{trans}}$ is the transformer module, $V$ denotes the vocabulary size, $d$ is the original token embedding dimension, and $l$ is the token number of the input text. For simplicity, the dimensions for each mode of the tensor and TT rank are represented as $I$ and $r$, respectively, which yields the highest compression ratio when $r = 1$ and $I = 2$ (as proved in Appx. D).

| **Memory** | Original | Compressed | Compressed Encoded Texts | Input to $\mathcal{M}_{\texttt{trans}}$ |
|---|---|---|---|---|
| | $\mathcal{O}(Vd)$ | $\mathcal{O}(VNIr^2)$ | $\mathcal{O}(lNIr^2)$ | $\mathcal{O}(ld)$ |
| **Computation** | TT-SVD | | Reconstruction | |
| | $\mathcal{O}(NIr^3)$ | | $\mathcal{O}(NIr^2)$ | |

Frobenius norm. Although the embedding vector is reshaped into a tensor, the decomposition for each mode of this tensor is still based on the matrix-level SVD (line 2). Then the complexity of TT_SVD can be derived from SVD and its variants, such as truncated SVD (Oseledets, 2011). Given the vocabulary size $V$, the original parameters of the embedding layers are compressed from $Vd$ to $V \sum_{k=1}^{N} r_{k-1} I_k r_k$, and the compression ratio can be obtained via $\eta_{\texttt{TTD}} = \frac{d}{\sum_{k=1}^{N} r_{k-1} I_k r_k} - 1$. The computation and memory complexities for all the above processes are summarized in Tab. 3.

**Energy Consumption Analysis.** Recall in Sec. 2.2 we have Rem. 1 to guide the choice between memory and computation for the same functionalities from the perspective of energy cost. Based on Rem. 1 and Tab. 3, we can initially give the estimated energy costs when the SLM processes an input token (only before the decoder). Assuming in the same operating environment and other conditions (e.g. temperature), the memory energy cost of each `float32` is $\nu$, and the computation energy cost of each `float32` is $\tau$, all the model weights are represented in `float32`.

When inputting a text of length $l$, denote original model energy cost regarding memory as $\mathcal{E}_\nu$, model energy cost regarding computation is $\mathcal{E}_\tau$,

$$\mathcal{E}_\nu = \nu(dV + ld), \quad \mathcal{E}_\tau = 0, \tag{2}$$

and after compressing with TensorSLM, the energy costs are

$$\mathcal{E}'_\nu = \nu(VNIr^2 + lNIr^2 + ld), \quad \mathcal{E}'_\tau = \tau NIr^2. \tag{3}$$

Denote the SVD rank $k$, the energy cost after compressing with matrix-based SVD is

$$\mathcal{E}''_\nu = \nu\left[k(V + 2d + l + 1) + ld\right], \quad \mathcal{E}''_\tau = \tau(2ldk - ld + kd). \tag{4}$$

Therefore, we have the ratio of inference energy $\omega$, between the compressed language models and the uncompressed models. Denote $\omega_{\texttt{TT}} = \frac{\mathcal{E}'_\nu + \mathcal{E}'_\tau}{\mathcal{E}_\nu + \mathcal{E}_\tau}$ as the ratio with TensorSLM, and $\omega_{\texttt{SVD}} = \frac{\mathcal{E}''_\nu + \mathcal{E}''_\tau}{\mathcal{E}_\nu + \mathcal{E}_\tau}$ as the ratio with SVD. We will give the estimated values of $\omega_{\texttt{TT}}$ and $\omega_{\texttt{SVD}}$ in Sec. 6.3 according to the hyperparameters of the investigated open-source SLMs.

## 5.2 Language Model Inference Process with the Compressed Embeddings

The original inference process with embedding vectors is as follows: when the encoded texts (separated as tokens) are forwarded to the embedding layer, the embedding layer outputs the embedding vectors according to the input tokens; the embedding layer here acts like a look-up table. The embedding vectors are then forwarded to the hidden layers of the transformer, whose size is the same as the dimension of the embedding vectors. Thus, if there is no internal change in the hidden layers, the dimension of the embedding vectors should compile with the dimension of the hidden layers. The compressed embeddings should be reconstructed to the original dimension to enable the forwarding process. This inference happens at the application phase shown in the upper right of Fig. 2.

Thus just before forwarding embedding vectors to the hidden layers, the memory usage increases from $l \sum_{k=1}^{N} r_{k-1} I_k r_k$ to $ld$. However, given that the vocabulary size $V$ is normally much larger than the input token number $l$, that means $V \gg l$. Thus our approach can still significantly reduce the memory usage if the embedding layer takes a significant part of the whole model parameters. The reconstruction process follows the tensor contraction in Eq. (7), turning the TT cores $\{\mathcal{G}^{(k)}\}$ into a $N$-order tensor $\mathcal{X}$ according to Eq. (1), and then vectorizing $\mathcal{X}$ into a full-size embedding vector according to Appx. A.2.

## 6    EMPIRICAL EVALUATION

### 6.1    EXPERIMENTAL SETUP

**Models, Tasks and Dataset.**    The sub-billion models we used are DistilGPT2 (Sanh, 2019), GPT2, GPT2-M/L (Radford et al., 2019), CerebrasGPT-111M/256M/590M (Dey et al., 2023), OPT-125M. We also tested the models of slightly over a billion parameters for language task performance with GPT2-XL (1.5 billion parameters), CerebrasGPT-1.3B and OPT-1.3B for the boundary tests.

Regarding the language tasks, we have two different level language tasks:

- **Simple Tasks**: language modelling and sentiment classification. For language modelling, the considered datasets are WikiText2, WikiText103 (Merity et al., 2017) and 1BW (Chelba et al., 2013). For sentiment classification, the considered dataset is IMDB (Maas et al., 2011).

- **Complex Tasks**: zero-shot common sense reasoning tasks.  The common sense reasoning datasets include ARC-easy and ARC-challenge Clark et al. (2018), BoolQ Clark et al. (2019), HellaSwag Zellers et al. (2019), PIQA Bisk et al. (2020), SIQA Sap et al. (2019) and Wino-Grade Sakaguchi et al. (2021).

**Hardware.**    Our main experiments were completed on a GPU workstation with an RTX A6000 48GB GPU and AMD Ryzen Threadripper PRO 5955WX CPU. The GPU resource was mainly used to fine-tune language modelling models for sequence classification, which is the requirement of the sentiment classification task. The inference latency of the low-end devices was measured on a Raspberry Pi 5, with a 64-bit Arm Cortex-A76 CPU and 8GB DRAM.

### 6.2    EVALUATION METRICS

**Compression Ratio.**    Denote $\mathcal{M}$ as a model block set containing a list of model modules like embedding layers and attention layers. With $\mathcal{M}_0$ as the original model block set, $\mathcal{M}_{\texttt{cmpr}}$ as the compressed version of $\mathcal{M}_0$, and $|\mathcal{M}|$ as the parameter count of $\mathcal{M}$.  The compression ratio $\eta$ is defined as

$$\eta = \frac{|\mathcal{M}_0| - |\mathcal{M}_{\texttt{cmpr}}|}{|\mathcal{M}_{\texttt{cmpr}}|}. \tag{5}$$

Specifically, the embedding compression rate is $\eta_{\texttt{emb}} = \frac{|\mathcal{T}_0| - |\mathcal{T}_{\texttt{cmpr}}|}{|\mathcal{T}_0|}$, where $\mathcal{T}$ only contains token embedding layer and position embedding layer.

**Perplexity and Logarithmic Perplexity.**    We use perplexity (`PPL`) as our metrics of language modelling. Furthermore, we use the logarithmic form of perplexity ($\ln$ `PPL` ) and its change ($\Delta \ln$ `PPL`) to align with the linearity of the compression ratio Eq. (5), as defined in Appx. C.

**Accuracy, Precision, Recall and F1-Score.**    We use these four common evaluation metrics for classification to analyze the classification performance of the compressed model comprehensively. To investigate the performance change before and after compression, we use the difference between the metric values after and before the compression.

**Zero-shot Reasoning Scores.**    For the metrics of reasoning tasks, we use the scores from (Clark et al., 2018; 2019; Zellers et al., 2019; Bisk et al., 2020; Sap et al., 2019; Sakaguchi et al., 2021).

**Energy Consumption.**    Since the actual energy consumption depends on multiple uncontrollable factors, as we discussed in Sec. 2.2, it is difficult to isolate compression energy cost from the actual measurements.  Thus, we use similar approaches in (Luo & Sun, 2024) to estimate the energy consumption.

We use the notations in Tab. 3 and Eq. (2) to (4), and approximate the ratio between computation energy cost and memory energy cost per `fload32` data as $\frac{\nu}{\tau} = 5$. Then, we got the configurations of the current open-source SLMs for the values of $d$, $V$ in  Eq. (2) to (4). Though we cannot get the actual energy costs, we can compare the inference energy costs of compressed and uncompressed models with this approach.

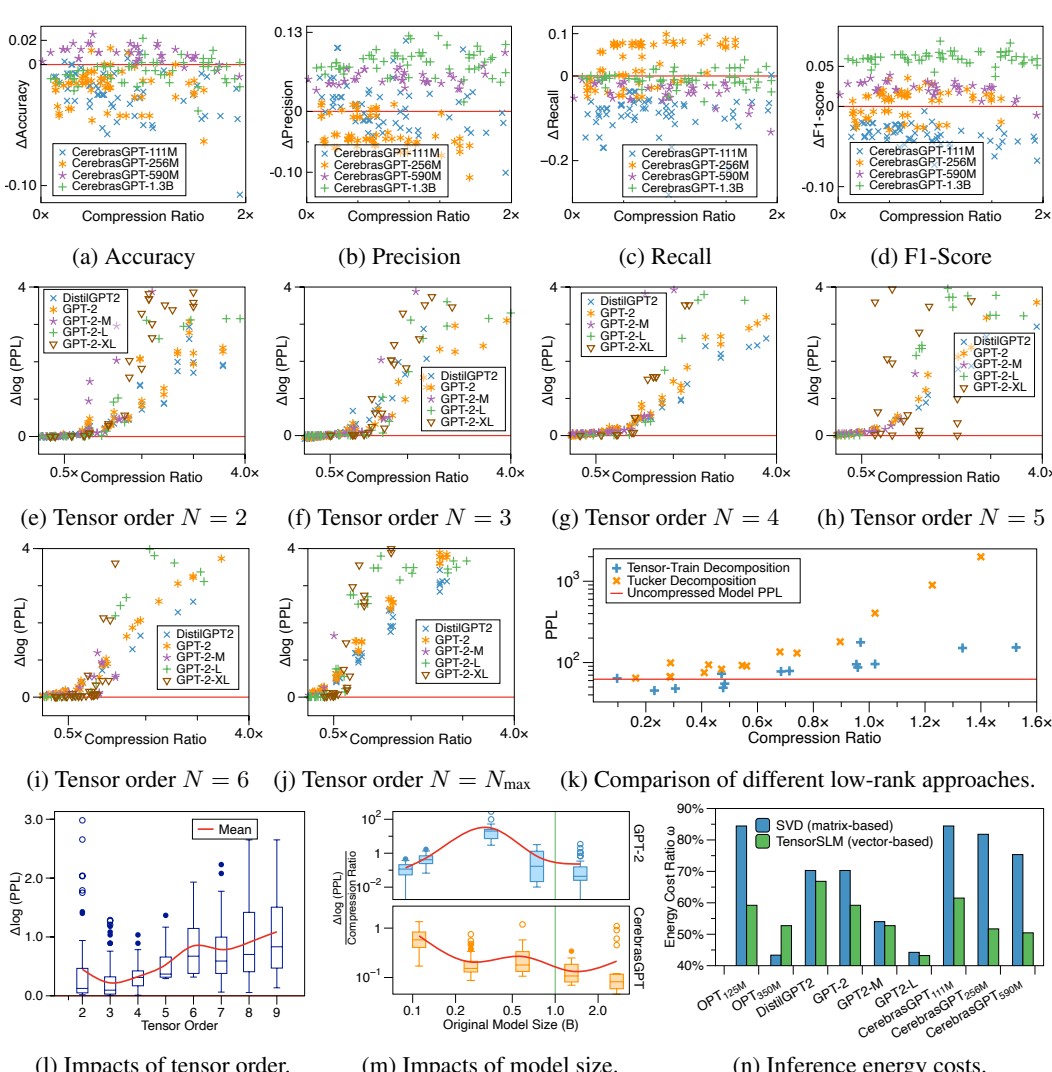

Figure 3: Experiments results. (a)-(d): Task performance of sentiment classification, with the increasing compression ratio. The higher the values, the better the classification performance. (e)-(j): The language modelling performance and compression ratio, where $N$ denotes the tensor order. (k): Different tensor decomposition approaches. (l) Task performance changes when the compression ratio is within $3.0\times$. When the tensor order increases, the language task performance tends to improve first, then decrease after order 3. (m) The compression trade-off on different sizes. This trade-off is roughly measured by the ratio between perplexity and compression ratio of embedding layers; the lower the ratio value, the better the compression. We found that the larger the model size, the better the trade-off, where CerebrasGPT has a smoother trend compared with GPT-2. (n) Energy costs ratio of compressed/uncompressed models, our approach overall outperform the SVD-based approach.

## 6.3 EXPERIMENTAL RESULTS

### 6.3.1 COMPRESSION RATIO AND LANGUAGE TASK PERFORMANCE.

**Language Modelling.** The language modelling performance and the compression ratio of different tensor orders and models are shown in Fig. 3. There is no general conclusion as to whether higher-order tensors are better than lower-order tensors, but roughly speaking, compression ratio and language modelling performance in Fig. 3e-Fig. 3h are better than those in Fig. 3i-Fig. 3j. This implies that there may exist a high-dimensional representation in the weight of embedding layers, no more than six-dimensional and most probably around three-dimensional. The best compression

cases (higher compression ratio with negligible drop in accuracy) occur when $N = 3$ (also as shown in Fig. 3l and 3m), implying the optimal feature representation of the token embedding vectors may be three-dimensional. On the other hand, due to the combination of tensor size and TT ranks exponentially exploding, we could not test all the possible combinations. However, we can still observe that independent of the tensor orders and the models used for the compression, significant language modelling performance loss tends to appear when the compression ratio exceeds $2.0\times$. We further compared our proposed approach with the Tucker decomposition in Fig. 3k with the same tensorization strategy in Sec. 5.1, and found our adopted Tensor-Train Decomposition outperforms the Tucker Decomposition in perplexity.

**Sentiment Classification.** The results of the sentiment classification task also show that the robustness of larger-scale models (Cerebras-590M and Cerebras-1.3B) is better than that of the smaller models (Cerebras-111M and Cerebras-256M), similar to the trend in language modelling tasks mentioned above. The compressed larger-scale models tend to outperform the original model in precision and F1-score, indicating that our compression improves the ability of the larger models to recognize the positive texts. In contrast, the smaller models tends to have worse performance when the compression ratio increases.

A notable observation is that in both language modelling (Fig. 3e to 3j) and sentiment classification (Fig. 3a to 3d), the larger models (GPT-2-M, GPT-2-L, GPT-2-XL, CerebrasGPT-590M and CerebrasGPT-1.3B) are more robust to the compression ratio increase, compared with smaller models (DistilGPT2, GPT-2, CerebrasGPT-111M and CerebrasGPT-256M), especially when the compression ratio is less than $1.0\times$. This is probably because the embedding layers take a smaller proportion of the entire model, and it also sheds light on the fact that TensorSLM might be used to improve the language task performance for large-scale models.

**Zero-shot Reasoning.** Since SLMs are incapable of the tasks that are too complex, we only evaluate the relatively simple reasoning tasks (e.g. those that do not involve multi-hop questioning, mathematics or multilingual), and the results are shown in in Appx. E. The bold numbers are the cases that outperform the uncompressed models, or the best in all the compressed cases.

Our approach has a higher chance of achieving better average reasoning task scores than the SVD-based approach, which implies that our tensors are better at extracting implicit representations in small-size models than matrices. Moreover, in our evaluation, our approach generally has higher scores than the SVD-based approach in ARC-challenge and BoolQ. Both of these datasets are more unprompted and unconstrained compared to the other evaluated datasets. This fact implies that our approach may be better at these difficult, unconstrained reasoning tasks.

### 6.3.2 LATENCY.

While TensorSLM significantly reduces the model parameters and even improves the language tasks performance, in practice it also introduced latencies - compression latency in Sec. 5.1 and inference latency Sec. 5.2.

For the compression latency, we investigated the compression latency on the token level, as shown in Tab. 4. Here, "original" means the uncompressed model, while $\mathtt{PPL}_\alpha$ means the compressed model with a negligible task performance drop. In our case, "negligible task performance drop" means in the language modelling task, the perplexity is no more than $100.0$. The notation $\varphi_{\max}$ refers to the compressed model with maximum compression ratio. We observed that for individual token embeddings, there was no significant latency difference between high-end servers and Raspberry Pi, typically no more than 2 milliseconds for each token. Thus, it is acceptable for the Raspberry Pi to compress the individual token embeddings.

For the inference latency of a single text, we chose a typical text length of 50 tokens, as shown in Tab. 5. we used "original", $\mathtt{PPL}_\alpha$, $\varphi_{\max}$ same as those in Tab. 4, to represent the uncompressed model, the compressed model with a negligible task performance drop and the model with a maximum compression ratio. A typically induced latency for an input text was no more than $0.3$ seconds, which is acceptable for edge applications.

Table 4: The latency (ms/token) of tensorization & decomposition token embedding vectors and reconstruction on the high-end and lower-end devices. $\mathtt{PPL}_\alpha$ means the compressed model with a negligible task performance drop, and the symbol $\varphi_{\max}$ represents the case with a maximum compression ratio. $d_{\mathrm{emb}}$ is the embedding dimension of the token embedding vector, and the tested models are GPT-2 and GPT-2-M. On the CPU level, for single token embedding vector decomposition and reconstruction, both server and edge devices have no significant computation overhead.

| Device (CPU) (ms/token) | $d_{\mathrm{emb}}$ | tensorization & decomposition | | reconstruction | |
|---|---|---|---|---|---|
| | | $\mathtt{PPL}_\alpha$ | $\varphi_{\max}$ | $\mathtt{PPL}_\alpha$ | $\varphi_{\max}$ |
| **Server** | 768 | 0.627 | 1.429 | 0.117 | 0.238 |
| | 1024 | 0.452 | 1.512 | 0.114 | 0.261 |
| **Raspberry Pi 5** | 768 | 0.760 | 1.948 | 0.330 | 0.468 |
| | 1024 | 0.612 | 2.148 | 0.364 | 0.614 |

Table 5: Parameters, number of floating-point operations (flops) of the compressed and uncompressed sub-billion models, and latency on Raspberry Pi CPU. For flops, the token number of the input texts is 100, while for latency on Raspberry Pi, the token number is 50.

| GPT Models | | GPT2 | | | | CerebrasGPT | | |
|---|---|---|---|---|---|---|---|---|
| | | DistilGPT2 | GPT-2 | GPT-2-M | GPT-2-L | 111M | 256M | 590M |
| **# Params** (M) | original | 81.9 | 124.44 | 354.82 | 774.03 | 111.05 | 255.98 | 590.31 |
| | $\mathtt{PPL}_\alpha$ | 67.06 | 106.36 | 326.45 | 734.28 | 101.78 | 226.69 | 543.45 |
| | $\varphi_{\max}$ | 43.45 | 85.99 | 303.88 | 710.83 | 71.87 | 200.59 | 511.07 |
| **flops** ($10^6$/text) | original | 20250 | 40490 | 142250 | 330980 | 14470 | 40400 | 103060 |
| | $\mathtt{PPL}_\alpha$ | +1.65 | +1.88 | +3.11 | +2.30 | +0.38 | +1.63 | +2.30 |
| | $\varphi_{\max}$ | +0.13 | +0.13 | +0.20 | +0.25 | +0.13 | +0.12 | +0.26 |
| **Latency on Raspberry Pi** (s/text) | original | $0.19_{\pm 0.02}$ | $0.50_{\pm 0.19}$ | $1.23_{\pm 0.12}$ | $3.01_{\pm 0.47}$ | $0.47_{\pm 0.21}$ | $0.71_{\pm 0.02}$ | $1.81_{\pm 0.25}$ |
| | $\mathtt{PPL}_\alpha$ | $0.36_{\pm 0.19}$ | $0.50_{\pm 0.16}$ | $1.26_{\pm 0.22}$ | $3.01_{\pm 0.29}$ | $0.48_{\pm 0.23}$ | $1.01_{\pm 0.29}$ | $1.89_{\pm 0.28}$ |
| | $\varphi_{\max}$ | $0.19_{\pm 0.03}$ | $0.71_{\pm 0.38}$ | $1.55_{\pm 0.36}$ | $3.52_{\pm 0.44}$ | $0.72_{\pm 0.42}$ | $0.95_{\pm 0.27}$ | $1.91_{\pm 0.24}$ |

### 6.3.3 ENERGY CONSUMPTION.

The estimated inference energy costs are shown in Fig. 3n. The Y-axis indicates the ratio between the inference energy costs of the compressed model and that of the uncompressed model; the lower, the better energy saving. For each language model, we select the compression case that has a similar language task performance according to Sec. 6.3.1.

We can observe that our approach is mostly better than the SVD-based approach. Furthermore, TensorSLM supports adaptivity in edge applications, while the SVD-based approach does not.

## 7 CONCLUSION AND FUTURE WORK

This paper focuses on the two unique requirements of Small Language Models (SLMs) deployed on low-end devices (i.e. Raspberry Pi) - *adaptivity* and *low energy*, and proposes a training-free approach to compress each token embedding based on Tensor-Train Decomposition. The proposed approach can cope with a dynamic environment by adjusting its vocabulary and "exchanging" memory with computation for longer battery life. The experimental evaluation covers GPT-2/CerebrasGPT and OPT series models, as well as both simple language modelling/classification and more complex zero-shot reasoning tasks. We also systematically measured the on-board inference latency of Raspberry Pi 5 and estimated the inference energy costs based on our rigorous analysis of computation and memory complexity. We found that the estimated inference energy cost is cut half off with neglectable inference latency and language task performance drop.

There are both limitations to our work and, more importantly, rather a broad range of future work following our proposed TensorSLM. Firstly, the tensorized embedding layers do not natively compile with the hidden layers, so tensorized hidden layers are required. Also, although tensor operations, like contraction, do not require much memory, they might need more arithmetic operations on the CPU, thus requiring accelerated tensor operations.

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

# Appendix

## A  PRELIMINARIES

### A.1  NOTATION

Table 6: Notation in this paper.

| Symbol | Meaning |
|---:|---|
| $a$ | Scalar |
| $\mathbf{x}$ | Vector |
| $\mathbf{A}$ | Matrix |
| $\mathcal{X}, \mathcal{A}, \mathcal{B}$ | Tensor |
| $N$ | Tensor order |
| $\mathcal{X}[i_1, i_2, \ldots, i_N]$ | The $(i_1, i_2, \ldots, i_N)$th entry of the tensor |
| $I, I_k$ | Tensor dimension, tensor dimension for the $k$th mode |
| $\mathcal{M}$ | Model module set |
| $|\mathcal{M}|, |\mathcal{G}|, |S|$ | Parameter count of the model module set $\mathcal{M}$, tensor $\mathcal{G}$ or cardinality of set $S$ |
| $V$ | Vocabulary of the language model |
| $d$ | Token embedding dimension |
| $l$ | Input text length |
| $r, r_k$ | TT rank, TT rank of the $k$th mode of the tensor |
| $\mathcal{G}^{(k)}$ | TT(MPS) core of the $k$th mode of the tensor |
| $\times_k^p$ | Tensor contraction for the $p$th (formal tensor) and $k$th (latter tensor) mode |
| $\eta$ | Compression ratio of the entire model |
| $\eta_{\text{emb}}$ | Compression ratio of the embedding layer |
| $\varphi$ | Parameter reduction ratio of the whole model. |
| $\varphi_{\text{emb}}$ | Parameter reduction ratio of the embedding layer. |
| $\nu$ | Memory energy consumption per `float32` data. |
| $\tau$ | Computation energy consumption per `float32` data. |
| $\mathcal{E}_\nu$ | Estimated energy cost regarding memory. |
| $\mathcal{E}_\tau$ | Estimated energy cost regarding computation. |
| $\omega_{\text{TT}}$ | Estimated energy cost ratio between the compressed model with TensorSLM and uncompressed model. |
| $\omega_{\text{SVD}}$ | Estimated energy cost ratio between the compressed model with SVD and the uncompressed model. |

---

**Algorithm 2:** Tensor-Train Singular Value Decomposition (TT-SVD)(Oseledets, 2011)

**Input**  : Data tensor, $\mathcal{X} \in \mathbb{R}^{I_1 \times I_2 \times \cdots \times I_N}$, and approximation accuracy, $\epsilon$

**Output** : Core tensors, $\mathcal{G}^{(1)}, \ldots, \mathcal{G}^{(N)}$, approximating $\mathcal{X} \in \mathbb{R}^{I_1 \times I_2 \times \cdots \times I_N}$

1  Initialize cores, $\mathcal{G}^{(1)}, \ldots, \mathcal{G}^{(N)}$, and $R_0 = 1$

2  Compute truncation parameter $\delta = \frac{\epsilon}{\sqrt{N-1}} \|\mathcal{X}\|_F$

3  $\mathcal{Z} \leftarrow \mathcal{X}$, and $\mathbf{Z} \leftarrow \mathbf{Z}_{(1)}$

4  **for** $n = 1$ *to* $N - 1$ **do**

5  $\quad$ Compute $\delta$-truncated SVD: $\mathbf{Z} = \mathbf{U}\mathbf{S}\mathbf{V} + \mathbf{E}$, s.t. $\|\mathbf{E}\|_F \leq \delta$; $\mathbf{U} \in \mathbb{R}^{R_{(n-1)}I_n \times R_n}$

6  $\quad$ $\mathcal{G}^{(n)} \leftarrow \texttt{reshape}\left(\mathbf{U}, [R_{(n-1)}, I_n, R_n]\right)$

7  $\quad$ $\mathbf{Z} \leftarrow \texttt{reshape}\left(\mathbf{S}\mathbf{V}^T, [R_n I_{(n+1)}, I_{(n+2)}I_{(n+3)} \ldots I_N])\right)$

8  $\mathcal{G}^{(N)} \leftarrow \mathbf{Z}$

9  **return** $\mathcal{G}^{(1)}, \mathcal{G}^{(2)}, \ldots, \mathcal{G}^{(N)}$

---

### A.2 TENSORS AND TENSOR OPERATIONS

This section gives brief mathematical preliminaries of tensor algebra, and basic knowledge in LLMs to facilitate the understanding of our proposed methodology in Sec. 5.

**Order-N Tensor.** An order-$N$ real-valued tensor is a multi-dimensional array, denoted by a calligraphic font, e.g., $\mathcal{A} \in \mathbb{R}^{I_1 \times \cdots \times I_N}$, where $N$ is the order of the tensor (i.e., number of modes), and $I_n$ $(1 \leq n \leq N)$ is the size (i.e., the dimension) of its $n$-th mode. Matrices (denoted by bold capital letters, e.g., $\mathbf{A} \in \mathbb{R}^{I_1 \times I_2}$) can be seen as order-2 tensors ($N = 2$), vectors (denoted by bold lower-case letters, e.g., $\mathbf{a} \in \mathbb{R}^I$) can be seen as order-1 tensors ($N = 1$), and scalars (denoted by lower-case letters, e.g., $a \in \mathbb{R}$) are order-0 tensors ($N = 0$).

**Tensor Entries.** The $(i_1, \ldots, i_N)$-th entry of an order-$N$ tensor is denoted by $a_{i_1, \cdots, i_N} \in \mathbb{R}$, where $i_n = 1, \ldots, I_n$ for $n = 1, \ldots, N$. A tensor fiber is a vector of tensor entries obtained by fixing all but one index of the original tensor (e.g., $\mathbf{a}_{:, i_2, i_3, \ldots, i_N} \in \mathbb{R}^{I_1}$). Similarly, a tensor slice is a matrix of tensor entries obtained by fixing all but two indices of the original tensor (e.g., $\mathbf{A}_{:, :, i_3, i_4, \ldots, i_N} \in \mathbb{R}^{I_1 \times I_2}$).

**Tensorization.** A vector $\mathbf{a} = (a_1, a_2, \ldots, a_{I_1 I_2 \cdots I_N}) \in \mathbb{R}^{I_1 I_2 \cdots I_N}$, can be tensorized (or "folded", "reshaped") into an order-$N$ tensor $\mathcal{A} \in \mathbb{R}^{I_1 \times I_2 \times \cdots \times I_N}$, so that

$$\mathcal{A}[i_1, i_2, \ldots, i_N] = a_{1 + \sum_{k=1}^{N} (i_k - 1) \prod_{p=1}^{k-1} I_p}, \qquad 1 \leq i_k \leq I_k, \tag{6}$$

where $\mathcal{A}[i_1, i_2, \ldots, i_N]$ denotes the $(i_1, i_2, \ldots, i_N)$-th entry of tensor $\mathcal{A}$.

**Vectorization.** Given an order-$N$ tensor, $\mathcal{A} \in \mathbb{R}^{I_1 \times \cdots \times I_N}$, its vectorization reshapes the high-dimensional matrix into a vector, $\text{vec}\,(\mathcal{A}) = \mathbf{a} \in \mathbb{R}^{I_1 \cdots I_N}$.

**Tensor Contraction.** The contraction of $\mathcal{A} \in \mathbb{R}^{I_1 \times \cdots \times I_N}$ and $\mathcal{B} \in \mathbb{R}^{J_1 \times \cdots \times J_M}$, over the $k$th and $p$th modes respectively, where $I_k = J_p$ is denoted as $\mathcal{A} \times_k^p \mathcal{B}$ and results in a tensor $\mathcal{C} \in \mathbb{R}^{I_1 \times \cdots \times I_{k-1} \times I_{k+1} \times \cdots \times I_N \times J_1 \times \cdots \times J_{p-1} \times J_{p+1} \times \cdots \times J_M}$, with entries

$$\mathcal{C}[i_1, \ldots, i_{k-1}, i_{k+1}, \ldots, i_N, j_1, \ldots, j_{p-1}, j_{p+1}, \ldots, j_M]$$
$$= \sum_{q=1}^{I_k} \mathcal{A}[i_1, \ldots, i_{k-1}, q, i_{k+1}, \ldots, i_N] \mathcal{B}[j_1, \ldots, j_{p-1}, q, j_{p+1}, \ldots, j_M] \tag{7}$$

**Matricization (Mode-n unfolding).** Mode-$n$ matricization of a tensor, $\text{mat}\,(\mathcal{A}, n) = \mathbf{A}_{\{n\}} \in \mathbb{R}^{I_n \times (I_1 \cdots I_{n-1} I_{n+1} \cdots I_N)}$, is a procedure of mapping the elements from a multidimensional array to a two-dimensional array (matrix). Conventionally, such procedure is associated with stacking mode-$n$ fibers (modal vectors) as column vectors of the resulting matrix. For instance, the mode-1 unfolding of $\mathcal{A} \in \mathbb{R}^{I_1 \times I_2 \times \cdots \times I_N}$ is represented as $\text{mat}\,(\mathcal{A}, 1) = \mathbf{A}_{\{1\}} \in \mathbb{R}^{I_1 \times (I_2 \cdots I_N)}$, where the subscript, $\{1\}$, denotes the mode of matricization, and is given by

$$\mathbf{A}_{(1)}\left[i_1, \overline{i_2 i_3 \ldots i_N}\right] = \mathcal{A}[i_1, i_2, \ldots, i_N] \tag{8}$$

Note that the overlined subscripts refer to linear indexing (or Little-Endian), given by:

$$\overline{i_1 i_2 \ldots i_N} = 1 + \sum_{n=1}^{N} \left[(i_n - 1) \prod_{n'=1}^{n-1} I_{n'}\right]$$
$$= 1 + i_1 + (i_2 - 1)I_1 + \cdots + (i_n - 1)I_1 \ldots I_{N-1} \tag{9}$$

### A.3 TENSOR-TRAIN SINGULAR VALUE DECOMPOSITION (TT-SVD)

Tensor-Train Singular Value Decomposition (TT-SVD) is clarified in Alg. 2.

# B    RELATED WORK IN DETAIL

Low-rank factorization can break the high-dimensional weight matrices into smaller matrices or tensors, so that the overall size of the model can be shrunk. According to the dimensions of the structure that the original weight matrices are broken into, these approaches can be divided into matrix-based and tensor-based.

**Matrix-based Approaches.**    A straightforward way to shrink the model size is to decompose weight matrices via singular value decomposition (SVD) (Acharya et al., 2019), which can be further improved by the weighted approach considering the model performance afterwards (Hsu et al., 2022), knowledge distillation (Lioutas et al., 2020; Mao et al., 2020) and pruning (Mao et al., 2020). There are also some block-wise decomposition approaches used in language model compression, like Kronecker Products (Tahaei et al., 2022; Edalati et al., 2022) and data-driven block-wise partitioning (Chen et al., 2018a; 2021).

(Dao et al., 2022; Qiu et al., 2024) used the block-diagonal matrices to reduce the FLOPs in the linear layers computation, with the bonus of shrinking the model size. However, our paper focuses on reducing the parameters of embedding layers, and there is no monotonous relationship between the FLOPs (computation cost) and parameters (memory usage) (Lin et al., 2020). Also, their investigated matrix multiplication only occurs in feed-forward layers, thus their approaches do not fit the embedding layer compression. Moreover, block-diagonal matrices are optimised for GPUs for better parallelization. Our aim of minimizing the number of parameters, makes it optimized for lower-end edge devices rather than GPUs. Indeed, on Raspberry Pi 5, the additional forwarding latency due to compressed embeddings (0.330 - 0.364ms /token in Tab. 4) is even faster than that on GPU (measured as 0.463ms /token in our setting), since there is no parallelization during this forwarding process.

**Tensor-based Approaches.**    Despite some efforts to use tensor decomposition to compress the language model size, all come with an extra training process. The works in (Abronin et al., 2024) use Kronecker decomposition with row-column permutation during the GPT model fine-tuning process, while (Hrinchuk et al., 2020) and (Chekalina et al., 2023b) propose a tensor-train structured embedding layer and GPT model respectively, yet both train the new-structured model from scratch.

# C    PERPLEXITY AND LOGARITHMIC PERPLEXITY.

Perplexity is used as a performance evaluation metric of the language modelling task, which has the following form

$$\mathrm{PPL}(S, \mathcal{M}) = \left( \prod_{i=1}^{|S|} p_{\mathcal{M}}(x_i | x_1, x_2, \ldots, x_{i-1}) \right)^{-1} \tag{10}$$

where $S$ is an ordered set (token sequence), consisting of a set of tokens $\{x_t\}$, $t = 1, 2, \ldots, |S|$, and $\mathcal{M}$ is the model block that contains all the modules of the language model we evaluate.

Notice that the compression ratio Eq. (5) has a linear form, while perplexity Eq. (10) has an exponential form, so it is hard to combine them as a description of a model compression result, since when compression ratio $\eta$ linearly increases, the perplexity $\mathrm{PPL}$ explodes exponentially. To this end, we use the following logarithmic form to describe the language modelling performance

$$\ln \mathrm{PPL}(S, \mathcal{M}) = - \sum_{i=1}^{|S|} \ln p_{\mathcal{M}}(x_i | x_1, x_2, \ldots, x_{i-1}) \tag{11}$$

Now, the language modelling performance change before and after compression is given by

$$\Delta \ln \mathrm{PPL}(S, \mathcal{M}) = \ln \mathrm{PPL}(S, \mathcal{M}_{\mathrm{cmpr}}) - \ln \mathrm{PPL}(S, \mathcal{M}_0) = \sum_{i=1}^{|S|} \ln \frac{p_{\mathcal{M}_0}(x_i | x_1, x_2, \ldots, x_{i-1})}{p_{\mathcal{M}_{\mathrm{cmpr}}}(x_i | x_1, x_2, \ldots, x_{i-1})}, \tag{12}$$

observe that Eq. (12) exhibits linearity, like Eq. (5).

# D PROOF OF THE HIGHEST COMPRESSION RATIO IN TAB. 3

Setting the tensor size $[I_1, \ldots, I_N]$ for the tensor $\mathcal{X}$ to achieve the highest compression rate, we next give the proof of this hyperparameter selection.

Regarding the definition of the compression rate in Sec. 6, and $r_0 = r_N = 1$ in Sec. 5.1, the compression rate can be represented as

$$\eta = \frac{V \times d}{\sum_{j=1}^{V} \sum_{n=1}^{N} (r_{n-1} \times I_n \times r_n)_j} \tag{13}$$

$$= \frac{V \times d}{I_1 r_1 + r_1 I_2 r_2 + \cdots + r_{N-2} I_{N-1} r_{N-1} + r_{N-1} I_N} \tag{14}$$

$$= \frac{V \times d}{\sum_{k=1}^{\lfloor \frac{N+1}{2} \rfloor} r_{2k-1} \left( r_{2k-2} I_{2k-1} + I_{2k} r_{2k+1} \right)} \tag{15}$$

For the simplest case, assume $I_1 = \cdots = I_N = I$ and $r_1 = \cdots = r_N = r$. Given $d = \prod_{n=1}^{N} I_n = I^N$, we have $N = \log_I D$, and

$$\eta = \frac{V \times d}{rI \left[ 2 + (N-2)r \right]} \tag{16}$$

$$= \frac{V \times d}{rI \left[ 2 + (\log_I d - 2) \right]}. \tag{17}$$

In Equation 17, the numerator is a constant, and in the denominator, $R$ is a hyperparameter for the Tensor-Train Decomposition. Thus the objective function for the highest compression rate $\eta$ is

$$\min_{I,N} rI \left[ 2 + (N-2) \right] \qquad \textbf{s.t.} \quad N = \log_I d \tag{18}$$

$$I, N, r \in \mathbb{Z}^+ \tag{19}$$

$$2 \le I \le N \le \lfloor \log_2 d \rfloor \tag{20}$$

Regarding Eq. (18), if eliminate $N$ then we have a function $h = rI \left[ 2 + (\log_I d - 2) \right]$. Regarding $d$ in Eq. (20), the largest token embedding size of recent GPT-3 (Brown, 2020) is 12,888. Thus, for the GPT series models no later than GPT-3, Eq. (18) should be $2 \le I \le N \le 13$. In this range, $h$ is a monotonically increasing function, where the minimum $h$ occurs at $I = 2$.

Therefore, for the simplest case, we have the best hyperparameter selection of $I_1 = I_2 = \cdots = I_N = 2$, $N = \lfloor \log_2 d \rfloor$.

# E    EXPERIMENT RESULTS OF ZERO-SHOT REASONING.

## E.1    OPT

The evaluation results of OPT-125M/350/1.3B as follows:

Table 7: Zero-shot reasoning performance for OPT-125M and its compressed versions.

|  | Param (%) | ARC-c | ARC-e | BoolQ | HellaS. | PIQA | SIQA | WinoG. | Avg. |
|---|---|---|---|---|---|---|---|---|---|
| **Original** | 100.00 | 23.38 | 57.11 | 57.74 | 41.53 | 71.71 | 34.49 | 59.35 | 49.33 |
| **SVD (matrices)** | 0.13 | **21.5** | 25.84 | 37.83 | 25.81 | 52.5 | 32.91 | 49.01 | 35.06 |
|  | 26.57 | 20.05 | 31.14 | 37.83 | 26.59 | 56.31 | 34.03 | 50.59 | 36.65 |
|  | 53.01 | 18.17 | 34.26 | 37.83 | 26.96 | 57.56 | 33.78 | **52.88** | 37.35 |
|  | 79.45 | 18.77 | 39.9 | 45.63 | 27.38 | 59.9 | **34.34** | 50.83 | 39.54 |
|  | 92.67 | 18.77 | **43.14** | 47.37 | 28.51 | **63** | 34.14 | 51.3 | 40.89 |
| **Ours (vectors)** | 2.47 | 21.33 | 26.39 | 37.83 | 25.63 | 52.94 | 33.98 | 50.59 | 35.53 |
|  | 29.17 | 20.22 | 28.66 | 39.14 | 26.17 | 53.54 | 33.83 | 49.88 | 35.92 |
|  | 50.78 | 21.25 | 29.55 | 40.15 | 26.19 | 54.9 | 33.37 | 50.12 | 36.50 |
|  | 71.88 | 19.37 | 35.31 | 47.09 | 27.78 | 59.58 | 33.37 | 50.59 | 39.01 |
|  | 87.11 | 19.03 | 39.6 | **59.51** | **28.41** | 61.15 | 33.78 | 51.14 | **41.80** |

Table 8: Zero-shot reasoning performance for OPT-350M and its compressed versions.

|  | Param (%) | ARC-c | ARC-e | BoolQ | HellaS. | PIQA | SIQA | WinoG. | Avg. |
|---|---|---|---|---|---|---|---|---|---|
| **Original** | 100.00 | 20.82 | 44.19 | 57.68 | 32.03 | 64.64 | 32.96 | 52.09 | 43.49 |
| **SVD (matrices)** | 0.20 | 21.5 | 25.25 | 37.83 | 25.67 | 51.36 | 32.32 | 49.17 | 34.87 |
|  | 19.93 | **20.82** | 25.93 | 38.53 | 25.94 | 53.92 | 32.11 | 51.07 | 35.62 |
|  | 39.66 | 20.22 | 25.8 | 38.62 | 26.22 | 53.26 | 32.16 | 50.59 | 35.41 |
|  | 59.39 | 19.2 | 25.55 | 38.5 | 26.54 | 53.97 | **33.03** | 51.46 | 35.61 |
|  | 79.12 | 19.2 | 27.53 | 37.83 | 27.16 | 55.93 | 32.62 | 49.33 | 35.80 |
|  | 98.85 | 20.73 | **41.37** | 37.89 | **30.47** | **62.95** | 32.73 | 49.25 | 39.48 |
| **Ours (vectors)** | 3.52 | 21.08 | 24.92 | 45.47 | 25.66 | 53.1 | **33.7** | 51.14 | 36.58 |
|  | 18.75 | 20.39 | 26.01 | **62.17** | 25.9 | 53.16 | 32.16 | 48.7 | 38.50 |
|  | 28.13 | 20.05 | 24.87 | **62.17** | 26.08 | 53.54 | **33.14** | 49.33 | 38.60 |
|  | 42.19 | 20.05 | 25.25 | 48.44 | 26.18 | 53.7 | 32.32 | 49.09 | 36.58 |
|  | 70.31 | 20.48 | 25.42 | **62.17** | 26 | 53.16 | 32.27 | 51.62 | 38.87 |
|  | 94.53 | **21.42** | 36.15 | 45.9 | 29.59 | 61.92 | **33.14** | **52.33** | **40.21** |

Table 9: Zero-shot reasoning performance for OPT-1.3B and its compressed versions.

| | Param (%) | ARC-c | ARC-e | BoolQ | HellaS. | PIQA | SIQA | WinoG. | Avg. |
|---|---|---|---|---|---|---|---|---|---|
| **Original** | 100.00 | 23.38 | 57.11 | 57.74 | 41.53 | 71.71 | 34.49 | 59.35 | 49.33 |
| **SVD (matrices)** | 0.05 | 21.93 | 26.43 | 38.75 | 25.66 | 53.75 | 33.32 | 49.8 | 35.66 |
| | 25.46 | 20.31 | 34.85 | 40.89 | 26.46 | 57.07 | 34.29 | 50.91 | 37.83 |
| | 50.87 | 20.56 | 44.87 | 57.34 | 28.04 | 63.28 | 35.52 | 51.3 | 42.99 |
| | 76.28 | 22.01 | 50.13 | **61.8** | 30.69 | 66.76 | **34.65** | 56.51 | 46.08 |
| | 96.61 | **23.55** | 53.96 | **60.28** | 36.35 | **69.59** | **34.7** | 57.77 | 48.03 |
| **Ours (vectors)** | 1.07 | 21.33 | 25.38 | 42.69 | 25.39 | 53.32 | 33.98 | 50.28 | 36.05 |
| | 24.22 | 21.16 | 25.93 | **60.43** | 25.88 | 54.9 | **34.54** | 50.36 | 39.03 |
| | 49.41 | 21.08 | 26.35 | 54.04 | 25.91 | 53.81 | **34.54** | 48.93 | 37.81 |
| | 70.70 | **25.26** | 52.86 | **57.98** | 38.78 | 69.48 | **35.31** | 58.48 | **48.31** |
| | 94.73 | **23.38** | **55.22** | 51.68 | **40.43** | 71 | **35.31** | **59.43** | 48.06 |

## E.2 CEREBRASGPT

The evaluation results of CerebrasGPT-111M/256M/590M/1.3B as follows:

Table 10: Zero-shot reasoning performance for CerebrasGPT-111M and its compressed versions

| | Param (%) | ARC-c | ARC-e | BoolQ | HellaS. | PIQA | SIQA | WinoG. | Avg. |
|---|---|---|---|---|---|---|---|---|---|
| **Original** | 100.00 | 16.64 | 37.88 | 62.14 | 26.76 | 59.41 | 33.88 | 49.01 | 40.82 |
| **SVD (matrices)** | 0.13 | 20.9 | 26.52 | 37.86 | 25.46 | 52.45 | 33.57 | 48.93 | 35.10 |
| | 26.57 | **17.49** | 31.44 | 37.77 | 26.53 | 56.2 | 33.57 | **50.99** | 36.28 |
| | 53.01 | **17.06** | 35.27 | 44.16 | 26.57 | 56.75 | 34.08 | **50.28** | 37.74 |
| | 79.45 | 16.55 | 37.08 | 59.88 | **26.76** | 58.81 | 33.88 | **49.72** | 40.38 |
| | 92.67 | 15.44 | **37.92** | 61.77 | **26.84** | **59.19** | 33.62 | **49.17** | **40.56** |
| **Ours (vectors)** | 2.47 | **19.97** | 28.24 | **61.93** | 26.09 | 54.13 | **34.54** | **50.36** | 39.32 |
| | 29.17 | **20.48** | 29.84 | 59.85 | 26.26 | 55.66 | **34.7** | 50.04 | 39.55 |
| | 50.78 | **19.8** | 31.48 | 49.11 | **26.78** | 57.51 | 33.42 | **49.09** | 38.17 |
| | 71.88 | **17.92** | 34.51 | 58.32 | 26.74 | 58.05 | **34.54** | **50.28** | 40.05 |
| | 87.11 | **20.99** | 24.07 | 61.04 | 25.66 | 52.67 | 33.98 | **49.49** | 38.27 |

Table 11: Zero-shot reasoning performance for CerebrasGPT-256M and its compressed versions

|  | Param (%) | ARC-c | ARC-e | BoolQ | HellaS. | PIQA | SIQA | WinoG. | Avg. |
|---|---|---|---|---|---|---|---|---|---|
| **Original** | 100.00 | 16.89 | 40.95 | 61.5 | 27.44 | 61.37 | 34.24 | 51.3 | 40.82 |
| **SVD (matrices)** | 0.09 | **21.16** | 26.73 | 37.83 | 25.74 | 52.29 | 32.8 | 51.22 | 35.40 |
|  | 28.26 | **17.75** | 33 | 38.2 | 26.48 | 58.05 | 33.98 | 50.43 | 36.84 |
|  | 47.05 | **17.15** | 35.61 | 39.97 | 26.83 | 59.03 | **34.19** | **51.78** | 37.79 |
|  | 75.22 | **18.09** | 39.44 | 61.01 | 27.4 | 60.83 | 34.03 | **51.62** | **41.77** |
|  | 94.00 | **18.17** | **40.74** | 59.94 | 27.4 | 61.04 | 33.67 | 50.91 | **41.70** |
| **Ours (vectors)** | 2.67 | **20.22** | 24.62 | 37.74 | 25.5 | 54.57 | **34.19** | 50.75 | 35.37 |
|  | 38.60 | **20.9** | 27.57 | 37.83 | 25.8 | 53.48 | 33.47 | 49.96 | 35.57 |
|  | 50.37 | **18.94** | 31.52 | 52.35 | 26.76 | 56.86 | 33.93 | 50.67 | 38.72 |
|  | 61.40 | **20.05** | 35.02 | 59.91 | 27.3 | 57.94 | 33.88 | 50.59 | 40.67 |
|  | 98.90 | **19.28** | **40.74** | **61.5** | **27.39** | **61.43** | 33.98 | **52.01** | **42.33** |

Table 12: Zero-shot reasoning performance for CerebrasGPT-590M and its compressed versions

|  | Param (%) | ARC-c | ARC-e | BoolQ | HellaS. | PIQA | SIQA | WinoG. | Avg. |
|---|---|---|---|---|---|---|---|---|---|
| **Original** | 100.00 | 19.03 | 46.42 | 59.17 | 29.12 | 62.73 | 35.31 | 49.8 | 43.08 |
| **SVD (matrices)** | 0.07 | **21.33** | 27.1 | 37.92 | 25.79 | 52.45 | 34.03 | 47.99 | 35.23 |
|  | 26.91 | 18.09 | 35.98 | 37.83 | 26.82 | 58 | 34.14 | **50.59** | 37.35 |
|  | 47.03 | 17.24 | 39.31 | 37.83 | 27.56 | 59.74 | 34.75 | **50.83** | 38.18 |
|  | 73.87 | 18.86 | 44.11 | 49.45 | 28.42 | 61.64 | 34.8 | **50.75** | 41.15 |
|  | 94.00 | **19.8** | **46.17** | 52.72 | 28.97 | **62.19** | **35.62** | **49.88** | 42.19 |
| **Ours (vectors)** | 1.37 | **23.38** | 24.87 | 56.61 | 25.59 | 52.67 | 33.73 | **52.17** | 38.43 |
|  | 19.21 | **19.97** | 26.81 | 49.82 | 25.66 | 52.72 | 34.24 | 49.17 | 36.91 |
|  | 46.88 | **19.54** | 35.9 | 40.89 | 27.04 | 57.83 | 34.54 | 49.72 | 37.92 |
|  | 66.41 | **20.31** | 38.09 | **58.87** | 28.21 | 60.28 | 34.29 | **49.88** | 41.42 |
|  | 94.34 | **22.1** | 44.7 | 56.42 | **29.02** | 61.64 | 35.52 | 49.49 | **42.70** |

Table 13: Zero-shot reasoning performance for CerebrasGPT-1.3B and its compressed versions

| | Param (%) | ARC-c | ARC-e | BoolQ | HellaS. | PIQA | SIQA | WinoG. | Avg. |
|---|---|---|---|---|---|---|---|---|---|
| **Original** | 100.00 | 22.35 | 50.88 | 59.33 | 32.55 | 66.49 | 34.44 | 51.93 | 45.42 |
| **SVD (matrices)** | 0.05 | 21.33 | 26.73 | 40.06 | 25.65 | 52.39 | 33.32 | 48.86 | 35.48 |
| | 25.46 | 18 | 38.59 | 37.83 | 27.38 | 59.3 | **34.95** | 51.85 | 38.27 |
| | 50.87 | 19.71 | 45.08 | 50.49 | 29.13 | 62.35 | 34.29 | **52.25** | 41.90 |
| | 76.28 | 19.97 | 49.03 | 55.6 | 30.7 | 64.85 | **34.49** | 49.96 | 43.51 |
| | 96.61 | 20.99 | **50.51** | 53.82 | 32.06 | **65.61** | **34.54** | 50.43 | 43.99 |
| **Ours (vectors)** | 1.07 | **22.53** | 27.95 | 40.03 | 25.86 | 54.35 | 34.24 | 50.75 | 36.53 |
| | 24.22 | 21.42 | 27.36 | 39.54 | 25.78 | 53.37 | 33.78 | 50.59 | 35.98 |
| | 57.03 | **22.61** | 39.56 | 53.52 | 30.44 | 63.06 | 33.62 | 47.75 | 41.51 |
| | 70.70 | **22.61** | 43.6 | **61.35** | 31.4 | 65.07 | **34.95** | 50.04 | **44.15** |
| | 94.73 | **22.35** | 46.97 | 56.36 | **32.21** | **65.61** | 33.98 | 51.78 | **44.18** |

