# OpenReview forum: "TensorGPT: Efficient Compression of Large Language Models based on Tensor-Train Decomposition"
_ICLR.cc/2025/Conference — Submitted to ICLR 2025_

### Official Review · Reviewer_cXwr · 2024-11-02

**Soundness:** 3
**Presentation:** 3
**Contribution:** 3
**Rating:** 6
**Confidence:** 3

**Summary:**

The paper presents TensorGPT, a novel approach for compressing LLMs through tensor-train decomposition of embedding layers. The key innovation is applying tensor-train decomposition to individual token embeddings without requiring additional training data or computation. The authors evaluate their method on GPT family models (GPT-2 and CerebrasGPT), demonstrating meaningful parameter reduction while maintaining or sometimes improving model performance. The work provides comprehensive evaluations on GPT-family models and demonstrates practical applicability on edge devices.

**Strengths:**

1. Novel training-free compression method specifically targeting embedding layers. Strong practical value for edge device deployment
2. Comprehensive experiments across multiple tasks and model sizes.  Solid theoretical foundation with clear mathematical derivations.
Thorough analysis of compression vs. performance trade-offs

**Weaknesses:**

1. Limited comparison with existing compression methods and baselines. The comparison would be more comprehensive by comparing with more baselines which are train-free and trained.
2. Evaluation focused mainly on GPT-family models and mainly focus on small models.
3. It would be a great to combine the proposed embedding compression method with other model compression methods to check compatibility.

**Questions:**

1. Could this approach be extended to show performance of other model architectures beyond the GPT family?
2. Could this approach compatible with other model compression methods?
3. Have you investigated the effect on model robustness like in the multilingual setting using more diverse tokens?

---

> ### Author Response · Authors · 2024-11-25
> **Thank you so much for your detailed comments and suggestions.**
>
> Thank you so much for your detailed comments and suggestions, our response is as follows.
>
> &nbsp;
> &nbsp;
>
> **Weakness 1. Limited comparison with existing compression methods and baselines.**
>
> >  The comparison would be more comprehensive by comparing with more baselines which are train-free and trained.
>
> Thanks for the suggestion. However, we think training is too heavy for low-end devices, since memory usage can be three to four times as much as inference [21].
>
> We have added SVD-based compression as our baseline in General Response G3.
>
> &nbsp;
> &nbsp;
>
>
> **Weakness 2. Evaluation is limited on small GPT models.**
>
> > Evaluation focused mainly on GPT-family models and mainly focus on small models.
>
> We have extended our experiments to OPT series models, please refer to General Response G3.
>
> Regarding the small model size, please refer to General Response G1.
>
> &nbsp;
> &nbsp;
>
>
> **Weakness 3. Compability with other model compression approaches.**
>
> > It would be a great to combine the proposed embedding compression method with other model compression methods to check compatibility.
>
> Thanks for the suggestion; we are considering quantization and plan to take it as future work.
>
> &nbsp;
> &nbsp;
>
> **Question 1. Language models outside GPT family.**
>
> We have extended to OPT, please refer to General Response G3.
>
> &nbsp;
> &nbsp;
>
> **Question 2. Could this approach compatible with other model compression methods?**
>
> Yes, quantization is the easiest one, and it is orthogonal to our approach.
>
> The combinations with weight tying mentioned by Reviewer JBLc, and pruning like that in SliceGPT[6] are a bit difficult, but we believe it can be solved by changing product/multiplication sequences.
>
> &nbsp;
> &nbsp;
>
> **Question 3. Have you investigated the effect on model robustness like in the multilingual setting using more diverse tokens?.**
>
> Thanks for the suggestion. At the moment, we suspect that the multilingual tasks are too complex for the small language models, as we discussed in Tab.xaFK.3, Weakness 3 with Reviewer xaFK.
> MGSM [19] in Tab.xaFK.3 is also a multilingual dataset, but all the investigated sub-billion models cannot perform well. We feel that it is better to evaluate robustness only when the language models perform well on the task.
>
> However, we acknowledge that multilingual tasks are rather valuable for the robustness of the edge application, and plan to take this as our future work.
> Yes, quantization is the easiest one and is orthogonal to our approach.
>
> &nbsp;
> &nbsp;
>
> **Reference**
>
> [19] "Language models are multilingual chain-of-thought reasoners." arXiv preprint arXiv:2210.03057 (2022).
>
> [21] Zhao, Jiawei, et al. "Galore: Memory-efficient llm training by gradient low-rank projection." arXiv preprint arXiv:2403.03507 (2024).

---

### Official Review · Reviewer_xaFK · 2024-11-03

**Soundness:** 2
**Presentation:** 3
**Contribution:** 2
**Rating:** 3
**Confidence:** 5

**Summary:**

The paper presents "TensorGPT," a model compression method using Tensor-Train Decomposition (TTD) to reduce the parameter size of Large Language Models (LLMs), particularly in embedding layers, without additional training. This training-free approach preserves model performance while significantly reducing memory requirements, demonstrated on low-end devices like Raspberry Pi.

**Strengths:**

1.	The approach does not require extra training, making it applicable for scenarios with limited resources or when extra data is unavailable.
2.	TensorGPT achieves substantial compression with low memory overhead, retaining performance across language modeling and sentiment analysis tasks.
3.	The paper includes experiments on low-end hardware (Raspberry Pi) and larger models, evaluating compression ratios, latency, and task-specific performance.

**Weaknesses:**

1.	The novelty is limited. Tensor-Train Decomposition is explored in several language model compression works. What are the differences between the proposed method and existing works?
2.	Lack important baselines. Compressing token embedding layers is studied in some papers and the paper does not compare with them. For example, [1] and baselines used in [1] should be set as baselines.
3.	The empirical evaluation is primarily on language modeling and sentiment classification, with potential limitations in representing other complex NLP tasks, such as math reasoning and multi-hop question answering.

[1] Wang, Haoyu, et al. "LightToken: A task and model-agnostic lightweight token embedding framework for pre-trained language models." Proceedings of the 29th ACM SIGKDD Conference on Knowledge Discovery and Data Mining. 2023.

**Questions:**

refer to weaknesses

---

> ### Author Response · Authors · 2024-11-25
> **Thank you so much for your detailed comments and reference.**
>
> Thank you so much for your comments, our response is as follows.
>
> &nbsp;
> &nbsp;
>
> **Weakness 1. The novelty is limited.**
>
> >  Tensor-Train Decomposition is explored in several language model compression works. What are the differences between the proposed method and existing works?
>
> As far as we know, [11,16,17] are the only works that use Tensor-Train Decomposition for the language model compression. All of them require extra training, which is unrealistic for low-end devices. Our training-free approach is suitable for meeting the requirements of adaptivity and low energy in edge applications, as we discussed in General Response G0.
>
> We would highly appreciate it if the Reviewer were aware of such work and possibly gave more references for it, so we could discuss them in our paper.
>
>
> &nbsp;
> &nbsp;
>
> **Weakness 2. Lack important baselines.**
>
> >  Compressing token embedding layers is studied in some papers and the paper does not compare with them. For example, [1] and baselines used in [1] should be set as baselines.
>
> Thanks for the reference. We have discussed the works about token embedding layer compression ([7,10-12, 14, 17]) in the General Response G2. Given that we only focus on small language models deployed on low-end devices, only [10] solves the same problem as ours and should be compared. However, [10] still requires training a meta-model, which should be fine-tuned when the deployment environment changes. So, we only take the SVD-based approach, the same as that in LightToken [7] (reference [1] in the Reviewer's original reviews), as our baseline. The experimental comparison of this part is in the General Response G3.
>
> &nbsp;
> &nbsp;
>
> **Weakness 3. Limited evaluation.**
>
> > The empirical evaluation is primarily on language modelling and sentiment classification, with potential limitations in representing other complex NLP tasks, such as math reasoning and multi-hop question answering.
>
> Thanks for pointing out. We report the results of zero-shot reasoning in General Response G3.
>
> We suspect that math reasoning and multi-hop question answering, which are still hard problems for LLMs, are too complex for sub-billion language models on low-end devices.
>
> We have evaluated some sub-billion language models on MGSM [19] for math reasoning and DROP [20] for multi-hop question answering, and it came out that all the models could not perform well. Our view is that there is no need to compress models for these two tasks on low-end devices.
>
> &nbsp;
> &nbsp;
>
>
> **Tab. xaFK.3 Small language model scores on math reasoning (MGSM) and multi-hop question answering (DROP)**
> | Task (metric or filter) |OPT-125M |OPT-350M | Qwen2.5-0.5B | CerebrasGPT-111M | CerebrasGPT-256M|CerebrasGPT-590M | distilgpt2 | gpt2 | gpt2-medium |
> |-----|------:|------|-----:|------|---|-----:|---|-----:|-----:|
> | MGSM (flexible extract)| 0.63|0.93 | 5.77| 0.23| 0.7| 0.57 | 0.3| 0.6| 0.73|
> | MGSM (remove whitespace)| 0 | 0 | 0.23| 0| 0|0 |0| 0| 0|
> | DROP (EM)|0.09 | 0.35 |0.04| 0 | 0.07| 0.05| 0 |0.07 | 0.05|
> | DROP (F1)| 2.26| 2.83 | 0.98| 15.3| 1.88 |1.88 |1.39 |2.72| 3.69|
> | | | | | | | |
>
> &nbsp;
> &nbsp;
>
> **Reference**
>
> [7] "LightToken: A task and model-agnostic lightweight token embedding framework for pre-trained language models." Proceedings of the 29th ACM SIGKDD Conference on Knowledge Discovery and Data Mining. 2023.
>
> [10] "Direction is what you need: improving word embedding compression in large language models." arXiv preprint arXiv:2106.08181 (2021).
>
> [11] "Tensorized embedding layers for efficient model compression." arXiv preprint arXiv:1901.10787 (2019).
>
> [17] "Efficient gpt model pre-training using tensor train matrix representation." arXiv preprint arXiv:2306.02697 (2023).
>
> [19] "Language models are multilingual chain-of-thought reasoners." arXiv preprint arXiv:2210.03057 (2022).
>
> [20] "DROP: A reading comprehension benchmark requiring discrete reasoning over paragraphs." arXiv preprint arXiv:1903.00161 (2019).

---

### Official Review · Reviewer_qcT4 · 2024-11-03

**Soundness:** 2
**Presentation:** 2
**Contribution:** 1
**Rating:** 3
**Confidence:** 5

**Summary:**

The authors applied tensor train to an edge device application, and found we can use that to compress some NLP models. It reads to me that the author is simply applying a pre-existing, well-known tensor decomposition method on a problem. And I didn't see any new knowledge developed or presented (I might be wrong, so hopefully the authors could point out clearly what's the real technical contribution other than yet another application).

**Strengths:**

- Working on a problem that's important.

**Weaknesses:**

- The authors claim "- As far as we know, we are the first to compress LLMs with low-rank factorization, specifically", which is not true to me. The architecture is the same for most NLP models and thus pardon me I really don't know what's added. I think all the tensor-train stuff has been developed and applied before. It's not immediately obvious to me what's new conclusions or findings drawn from this paper.

**Questions:**

- Pardon me that I really don't know what's the biggest contribution. Other than your claim of try things on LLM, what's the real technical contribution in your paper not discussed by previous works?

---

> ### Author Response · Authors · 2024-11-25
> **Thanks for your comments.**
>
> Thanks for your comments, our response is as follows.
>
> &nbsp;
> &nbsp;
>
> **Weakness 1. Overclaim**
>
> >The authors claim "- As far as we know, we are the first to compress LLMs with low-rank factorization, specifically", which is not true to me.
>
> Thanks for pointing out. We acknowledge that it is confusing and misleading here. We wanted to convey "As far as we know, we are the first to compress Small Language Models (SLMs) [9] for **low-end devices** use cases, with low-rank factorization'', which has been updated in line 78-80 of the updated submission.
>
> > The architecture is the same for most NLP models and thus pardon me I really don't know what's added.
>
> We guess the Reviewer wanted to express that our approach has no obvious contribution in algorithms, since model compression is to work on the models with existing architectures rather than creating a new architecture.
>
> For a short answer, our approach is specifically designed for small language models on low-end devices, to meet the requirements of adaptivity and low energy in edge applications. We systematically analysed energy, latency, etc, which common LLM compression rarely considers.
>
> For a detailed answer, please refer to the General Response G0, G2 and our answers to Weakness 4 raised by Reviewer JBLc.
>
> &nbsp;
> &nbsp;
>
> **Weakness 2. About Tensor-Train**
>
> >I think all the tensor-train stuff has been developed and applied before.
>
> General Response G1 includes all the relevant tensor-train stuff [11,16,17] we know. All [11,16,17] contain extra training, which is unsuitable for low-end devices. In the updated submission, this part has been emphasized in line 189-193.
>
> We would highly appreciate it if the Reviewer were aware of such work and possibly gave more references for such work, so that we could discuss them in our paper.
>
> &nbsp;
> &nbsp;
>
> **Weakness 4. Conclusion**
>
> >It's not immediately obvious to me what's new conclusions or findings drawn from this paper.
>
> Conclusion: our approach is suitable for low-end devices in edge applications.
>
> &nbsp;
> &nbsp;
>
> **Question**
> >Pardon me that I really don't know what's the biggest contribution. Other than your claim of try things on LLM, what's the real technical contribution in your paper not discussed by previous works?
>
> As we mentioned in the response to Weakness 2, our contribution is the first work for low-end devices to design a low-rank model compression strategy to satisfy the adaptivity and low energy requirements in edge applications.
>
> For the comparison between relevant works and ours, please refer to General Response G2.
>
> &nbsp;
> &nbsp;
>
> **Reference**
>
> [8] "MobileLLM: Optimizing Sub-billion Parameter Language Models
> for On-Device Use Cases", ICML 2024
>
> [9] "Small Language Models: Survey, Measurements, and Insights." arXiv preprint arXiv:2409.15790 (2024).
>
> [11] "Tensorized embedding layers for efficient model compression." arXiv preprint arXiv:1901.10787 (2019).
>
> [17] "Efficient gpt model pre-training using tensor train matrix representation." arXiv preprint arXiv:2306.02697 (2023).

---

### Official Review · Reviewer_JBLc · 2024-11-06

**Soundness:** 2
**Presentation:** 3
**Contribution:** 2
**Rating:** 3
**Confidence:** 5

**Summary:**

This paper proposes using tensor-train decomposition to compress the token embedding matrix, aiming to reduce model size and accelerate inference, particularly in edge device scenarios. The technique leverages a higher-order tensor approximation method, built upon singular value decomposition (SVD), to efficiently represent embeddings while maintaining performance. This tailored approach is well-suited for resource-constrained environments, promising benefits in storage and computational efficiency.

**Strengths:**

The paper is well-structured and easy to follow, with a clear presentation of the methodology and its applications.

**Weaknesses:**

**Limited Novelty:** The method largely builds upon the existing TT_SVD approach, with Algorithm 1 in this paper replicating methods already established in prior literature [1]. The methodological advancements or generalizations beyond TT_SVD appear minimal based on the methodology section.

**Unconvincing Experiments:** The experiments are conducted on older and relatively small models, such as GPT-2 and models up to only 1.3B parameters. This limits the relevance of the results, as they don’t reflect performance on contemporary large language models (LLMs). Additionally, the evaluation setup lacks modern benchmarking practices; for instance, LLM harness [2] would provide a more standardized evaluation framework.

**Unintuitive Rationale:** The foundational rationale for embedding a matrix using a tensor is unclear. The method requires reshaping the matrix into a higher-order tensor before applying tensor decomposition, but the paper does not provide an intuitive explanation for why this approach is effective or reasonable. Also, the embedding matrix only occupies a small part of the model even for a mid size of model, and a common way to reduce parameter size is through weight tie, which directly reduce the parameter sizes of embedding into half. How does you method performs in this case?

**Overstatement of Contribution:** Some claims appear overstated. For instance, the authors state, "As far as we know, we are the first to compress LLMs with low-rank factorization, specifically for low-end devices." However, the cited reference [3] already demonstrates compression of LLMs using SVD for similar purposes. Furthermore, there are numerous existing works that apply low-rank decomposition for LLM compression, such as [4-6].

**Limited Impact on Overall Model Size:** The embedding matrix occupies only a small portion of the model's parameters, even in mid-sized models. A commonly used technique, weight tying, can directly halve the embedding parameter size, offering a straightforward compression approach. The paper does not address how the proposed method compares to weight tying or performs when weight tying is already applied, raising questions about its practical impact on overall model size reduction.

[1] V. Oseledets. Tensor-train decomposition. SIAM Journal on Scientific Computing, 33(5):2295–2317,
2011. doi: 10.1137/090752286.

[2] Gao, Leo, et al. "A framework for few-shot language model evaluation." Version v0. 0.1. Sept 10 (2021): 8-9.

[3] Yen-Chang Hsu, Ting Hua, Sungen Chang, Qian Lou, Yilin Shen, and Hongxia Jin. Language model
compression with weighted low-rank factorization. In International Conference on Learning
Representations, 2022.

[4] Yuan, Zhihang, et al. "Asvd: Activation-aware singular value decomposition for compressing large language models." arXiv preprint arXiv:2312.05821 (2023).

[5] Lin, Chi-Heng, et al. "MoDeGPT: Modular Decomposition for Large Language Model Compression." arXiv preprint arXiv:2408.09632 (2024).

[6] Ashkboos, Saleh, et al. "Slicegpt: Compress large language models by deleting rows and columns." arXiv preprint arXiv:2401.15024 (2024).

**Questions:**

How does your method perform with weight tying?
What is the percentage of overall compression for model sizes larger than 13B?

---

> ### Author Response · Authors · 2024-11-25
> **Thank you so much for the very detailed comments and references.**
>
> ## **(1/2)**
>
> Thank you so much for the very detailed comments and references! Our response is as follows.
>
> &nbsp;
> &nbsp;
>
> **Weakness 1. Limited Novelty**
> > ... The methodological advancements or generalizations beyond TT_SVD appear minimal based on the methodology section.
>
> Though we did not change the exact implementation of TT-SVD, we did adjust its working unit (vectors rather than the whole matrix) and workflow to cope with the unique issues  (adaptivity and low energy, as stated in General Response G0) for low-end devices. We also have a comprehensive systematic analysis regarding latency, energy and flops, and the impacts (latency and accuracy) of the tensor orders in the experimental section.
>
> Also, we do not believe a simple methodology necessarily means no novelty. "Attention is All You Need" and "LoRA: Low-rank Adaptation of Large Language Models" are two examples. Their methodologies are rather simple (as stated in their abstract or introduction), but both revolutionised the community.
>
> &nbsp;
> &nbsp;
>
> **Weakness 2. Unconvincing Experiments:**
>
> > The experiments are conducted on older and relatively small models, such as GPT-2 and models up to only 1.3B parameters...
>
> Please refer to General Response G1, we only focus on SLMs on low-end device.
>
> > ... the evaluation setup lacks modern benchmarking practices...LLM harness [2] ...
>
> Thank you so much for the helpful evaluation resources, our new experiment results are shown in General Response G3.
>
> &nbsp;
> &nbsp;
>
> **Weaknesses 3. Unintuitive Rationale:**
>
> > The foundational rationale for embedding a matrix using a tensor is unclear. The method requires reshaping the matrix into a higher-order tensor before applying tensor decomposition, but the paper does not provide an intuitive explanation for why this approach is effective or reasonable.
>
> Tensors can model implicit high-dimensional representations (as well as the interactions among orders) of the model weights. In this sense, tensors have better expressivity than matrices, which is also a good solution for small-size models to express complex functionality with limited parameter space.
>
> The results in G3 have empirically proved this point, as tensor-based approaches have higher chances of retaining the language task performance than the matrix-based approach.
>
> > ... the embedding matrix only occupies a small part of the model even for a mid size of model, and a common way to reduce parameter size is through weight tie, which directly reduce the parameter sizes of embedding into half. How does you method performs in this case?
>
> Firstly, apart from reasoning and language modelling, classification is also an important task for edge applications, which we have investigated in the paper. Weight tying cannot be compiled with the classification layer.
>
> Secondly, we acknowledge that weight tying is a common approach to reducing memory, but we do not reckon it is a necessary part of LMs. Especially for the adaptivity requirements in edge applications (Section 2.1 in the updated submission), there should be a different approach to compress the fully connected layers (i.e. further amplifying the signal from the transformer) rather than directly reusing the weights of the embedding layer.
>
> &nbsp;
> &nbsp;
>
> **Weakness 4. Overstatement of Contribution:**
>
> >  Some claims appear overstated. For instance, the authors state, "As far as we know, we are the first to compress LLMs with low-rank factorization, specifically for low-end devices."
>
> Thanks for pointing out. We acknowledge it is confusing and misleading. We wanted to convey "As far as we know, we are the first to compress Small Language Models (SLMs) for **low-end devices** use cases, with low-rank factorization'', which has been updated in line 82-83 of the updated submission.
>
> > However, the cited reference [3] already demonstrates the compression of LLMs using SVD for similar purposes. Furthermore, there are numerous existing works that apply low-rank decomposition for LLM compression, such as [4-6].
>
> Thanks for the references. [3-6] are discussed in General Response G2.
>
> Furthermore, we want to clarify the inconsistency of the term "LLMs". The use of the term "LLM'' in the original submission is to follow the term used in MobileLLM [8]. To avoid confusion, we emphasized that we only focus on **sub-billion parameter models** in the abstract and the introduction of our original submission. We have updated the paper title in our new submission accordingly, **TensorSLM: Sub-billion Parameter Language Model Compression for Low-end Devices based on Tensor-Train Decomposition**. This clarifies our focus is not "LLMs" but Small Language Models (SLMs) [9].

---

> > ### Author Response · Authors · 2024-11-25
> >
> > ## **(2/2)**
> >
> > &nbsp;
> > &nbsp;
> >
> > **Weakness 5. Limited Impact on Overall Model Size:**
> >
> > > The embedding matrix occupies only a small portion of the model's parameters, even in mid-sized models.
> >
> > Please refer to the General Response G1.
> >
> > > A commonly used technique, weight tying, can directly halve the embedding parameter size, offering a straightforward compression approach. The paper does not address how the proposed method compares to weight tying or performs when weight tying is already applied, raising questions about its practical impact on overall model size reduction.
> >
> > Please refer to the last response to Weakness 3.
> >
> > &nbsp;
> > &nbsp;
> >
> > **Question 1. How does your method perform with weight tying?**
> >
> > Please refer to the last response to Weakness 3.
> >
> > &nbsp;
> > &nbsp;
> >
> > **Question 2. What is the percentage of overall compression for model sizes larger than 13B?**
> >
> > Please refer to the General Response G1.
> >
> > &nbsp;
> > &nbsp;
> >
> > **Reference**
> >
> > [1] "Tensor-train decomposition." SIAM Journal on Scientific Computing, 33(5):2295–2317, 2011. doi: 10.1137/090752286.
> >
> > [2] "A framework for few-shot language model evaluation." Version v0. 0.1. Sept 10 (2021): 8-9.
> >
> > [3] "Language model compression with weighted low-rank factorization." ICLR 2022.
> >
> > [4] "Asvd: Activation-aware singular value decomposition for compressing large language models." arXiv preprint arXiv:2312.05821 (2023).
> >
> > [5] "MoDeGPT: Modular Decomposition for Large Language Model Compression." arXiv preprint arXiv:2408.09632 (2024).
> >
> > [6] "Slicegpt: Compress large language models by deleting rows and columns." arXiv preprint arXiv:2401.15024 (2024).
> >
> > [7] "LightToken: A task and model-agnostic lightweight token embedding framework for pre-trained language models." Proceedings of the 29th ACM SIGKDD Conference on Knowledge Discovery and Data Mining. 2023.
> >
> > [8] "MobileLLM: Optimizing Sub-billion Parameter Language Models
> > for On-Device Use Cases", ICML 2024
> >
> > [9] "Small Language Models: Survey, Measurements, and Insights." arXiv preprint arXiv:2409.15790 (2024).

---

> > ### Comment · Reviewer_JBLc · 2024-11-26
> >
> > Thank you for your response. However, the response did not fully address my concerns regarding two fundamental aspects:
> >
> > **Compression of small language models (SLMs) is less general than for large language models (LLMs):** LLMs are more widely used, and their compression techniques can often be applied to smaller models, whereas the reverse is not always true. For instance, methods like LLM-Pruner [1] and layer pruning strategies [2] demonstrate fast and effective compression, even for large models.
> >
> > **Lack of comparisons with state-of-the-art (SOTA) methods:** The additional comparisons with SVD are insufficient to justify the proposed method. SVD performs significantly worse compared to other SOTA approaches, making it an inadequate baseline.
> >
> > In summary, while the authors have clarified the paper's applicability, I believe the method lacks sufficient generality and empirical support to justify its contribution for publication in this venue. Therefore, I will retain my score.
> >
> > [1] Ma, Xinyin, Gongfan Fang, and Xinchao Wang. "LLM-Pruner: On the Structural Pruning of Large Language Models." Version v1. May 19 (2023). arXiv:2305.11627.
> >
> > [2] Men, Xin, et al. "ShortGPT: Layers in Large Language Models are More Redundant Than You Expect." Version v1. March 6 (2024). arXiv:2403.03853.

---

> ### Author Response · Authors · 2024-12-02
> **Thank you so much for your reply and new references**
>
> # (1/3)
> Thank you so much for your reply and new references for LLM-Pruner [22] and ShortGPT [23]. Our responses to your reply are as follows:
>
> &nbsp;
> &nbsp;
> &nbsp;
> &nbsp;
>
> **1. Compression of small language models (SLMs) is less general than for large language models (LLMs).**
>
> > LLMs are more widely used, ...
>
> We acknowledge that "LLMs are more widely used" at the moment, but we do think SLM compression is also important. SLMs are for on-device applications (e.g. mobile phones and Raspberry Pi), and are suitable for applications without stable networks, sufficient GPU resources or continuous power charging. The compression of SLMs directly impacts battery life and, therefore, the user experience.
>
> > ... and their compression techniques can often be applied to smaller models, whereas the reverse is not always true.
>
> There has yet to be a consensus that the compression approaches of LLMs can be easily migrated to SLMs.
> On the contrary, there exists empirical evidence that LLM compression approaches cannot maintain the accuracy of SLMs as they can for LLMs:
>
> 1. In Tab.2 of ShortGPT [23], for the same model series (Llama2 and Baichuan2), 13B models maintain more model accuracy than 7B models with ShortGPT, as summarized as follows
>
>
> **Tab.JBLc.1.  Average score degradation after compression with ShortGPT.**
> |  $\Delta$ Avg.        | 7B    | 13B   |
> |-----------|-------|-------|
> | Llama2    | **-6.54** | -4.61 |
> | Baichuan2 | **-8.59** | -7.88 |
> | | | |
>
>
> 2. As stated in the second last paragraph of the introduction of SparseGPT [24], **"larger models are more compressible"**. In Fig.2 of [24], with the same compression settings, the models with fewer parameters have a more severe accuracy drop (larger slope in Fig.2).
>
> 3. In line 418-420 and line 445-448 of our current submission, we addressed that our compression approach performs better on the larger-sized models, which can be observed from Fig. 3 (a-b,d-j,m) in our current submission.
>
>
> > For instance, methods like LLM-Pruner [1] and layer pruning strategies ShortGPT [2] demonstrate fast and effective compression, ...
>
> Thanks for this information. In the referred LLM-Pruner (7B as their smallest tested models) and ShortGPT (2.8B as their smallest tested models) in detail, we did not find their results on sub-billion parameter models.
> As we stated in line 58 of our current submission, running an uncompressed Gemma-2B on Raspberry Pi leads to a system crash. Thus we only consider models around or less than 1B.
>
> > ... , even for large models.
>
> Our paper only focuses on SLMs running on **low-end devices**, which normally cannot hold LLMs.
> Thus we do not consider the "large models".

---

> ### Author Response · Authors · 2024-12-02
>
> # (2/3)
> &nbsp;
> &nbsp;
>
> **2. Lack of comparisons with state-of-the-art (SOTA) methods.**
>
> >The additional comparisons with SVD are insufficient to justify the proposed method. SVD performs significantly worse compared to other SOTA approaches, making it an inadequate baseline.
>
> Thanks for this emphasis. To find an appropriate baseline, we investigated the references in our General Response, the newly referred LLM-Pruner[22], ShortGPT[23] and a commonly used baseline SparseGPT[24]. Among these references, only  SliceGPT[6] is **training-free** and **compresses the embedding layers**. The comparisons are in Tab.JBLc.2 and Tab.JBLc.3. Given that SliceGPT also compresses other layers, we only listed the results of similar overall parameter ratios after the compression. The **bold** numbers are the best performance for each parameter ratio setting.
>
>
>
> **Tab.JBLc.2. Zero-shot performance of OPT-125M after compression.**
> |  OPT-125M            | **Params %** | **ARC-c** | **ARC-e** | **BoolQ** | **HellaS.** | **PIQA**  | **WinoG.** | **Avg.**  |
> |--------------|--------------|-----------|-----------|-----------|-------------|-----------|------------|-----------|
> | Original| 100 | 23.38 | 57.11 | 57.74 | 41.53 | 71.71 |  59.35 | 50.91|
> | **SparseGPT 2:4**|  -       | 19.03     | 37.12 | 58.59     | 27.77   | 58.32     | 51.7  | 42.09|
> | **SVD**    | 77.36        | 20.05     | 31.14     | 37.83     | 26.59       | **56.31** | 50.59      | 37.09     |
> |              | 85.51        | 18.17     | 34.26     | 37.83     | 26.96       | 57.56     | **52.88**  | 37.94     |
> |              | 97.74        | 18.77     | **43.14** | **47.37** | 28.51       | **63.00** | **51.30**  | **42.02** |
> |              |              |           |           |           |             |           |            |           |
> | **SliceGPT**| 77.15        | 19.20     | **35.14** | 37.86     | **27.38**   | 55.33     | **51.93**  | **37.81** |
> |              | 86.20        | 19.11     | **38.55** | 37.92     | **28.04**   | **58.00** | 50.20      | **38.64** |
> |              | 99.16        | **20.39** | 41.46     | 40.00     | 28.84       | 61.59     | 50.28      | 40.43     |
> |              |              |           |           |           |             |           |            |           |
> | **Ours**     | 78.16        | **20.22** | 28.66     | **39.14** | 26.17       | 53.54     | 49.88      | 36.27     |
> |              | 84.83        | **21.25** | 29.55     | **40.15** | 26.19       | 54.90     | 50.12      | 37.03     |
> |              | 99.76        | 20.05     | 38.68     | 45.41     | **28.86**   | 61.53     | 49.88      | 40.74     |
> | ||

---

> ### Author Response · Authors · 2024-12-02
>
> # (3/3)
> &nbsp;
> &nbsp;
>
> **Tab.JBLc.3. Zero-shot performance of OPT-1.3B after compression.**
>
> | OPT-1.3B             | **Params %** | **ARC-c** | **ARC-e** | **BoolQ** | **HellaS.** | **PIQA**  | **WinoG.** | **Avg.** |
> |--------------|--------------|-----------|-----------|-----------|-------------|-----------|------------|----------|
> | Original|100 | 22.35| 50.88|59.33 | 32.55| 66.49| 51.93| 47.26 |
> | **SparseGPT 2:4**|  -       | 19.88     | 44.82 | 57.34     | 32.97   | 63.49     | 55.49  | 45.67|
> | **SVD**      | 96.16        | 20.56     | 44.87     | **57.34** | 28.04       | 63.28     | 51.30      | 44.23    |
> |              | 98.14        | 22.01     | 50.13     | **61.80** | 30.69       | 66.76     | 56.51      | 47.98    |
> |              | 99.73        | 23.55     | **53.96** | **60.28** | 36.35       | 69.59     | 57.77      | **50.25**    |
> |              |              |           |           |           |             |           |            |          |
> | **SliceGPT** | 96.29        | **24.15** | **53.66** | 46.91     | **37.18**   | **67.46** | **55.41**  | **47.46**    |
> |              | 97.81        | 24.15     | **55.39** | 47.95     | **39.08**   | 68.72     | 56.75      | 48.67    |
> |              | 99.91        | **23.72** | 55.22     | 48.13     | 38.34       | 68.44     | 55.72      | 48.26    |
> |              |              |           |           |           |             |           |            |          |
> | **Ours**     | 96.04        | 21.08     | 26.35     | 54.04     | 25.91       | 53.81     | 48.93      | 38.35    |
> |              | 97.71        | **25.26** | 52.86     | 57.98     | 38.78       | **69.48** | **58.48**  | **50.47**    |
> |              | 99.59        | 23.38     | 55.22     | 51.68     | **40.43**   | **71**    | **59.43**  | 50.19    |
> |||
>
> &nbsp;
> &nbsp;
> &nbsp;
> &nbsp;
>
> We also give the results of SparseGPT. Since SparseGPT only freezes the weights rather than "deletes" them, we do not list its parameter ratio. From Tab Tab.JBLc.2 and Tab.JBLc.3, we can observe that the different compression approaches have different superiorities for different zero-shot reasoning tasks and compression ratios. Even SVD-based sometimes outperforms the others.
>
> These results further indicate that *compression approaches of LLMs may not be easily migrated to SLMs*, as we discussed in (1/3).
>
> **References**
>
> [6] "Slicegpt: Compress large language models by deleting rows and columns." arXiv preprint arXiv:2401.15024 (2024).
>
> [22] Ma, Xinyin, Gongfan Fang, and Xinchao Wang. "LLM-Pruner: On the Structural Pruning of Large Language Models." Version v1. May 19 (2023). arXiv:2305.11627.
>
> [23] Men, Xin, et al. "ShortGPT: Layers in Large Language Models are More Redundant Than You Expect." Version v1. March 6 (2024). arXiv:2403.03853.
>
> [24] Frantar, Elias, and Dan Alistarh. "Sparsegpt: Massive language models can be accurately pruned in one-shot." International Conference on Machine Learning. PMLR, 2023.

---

### Author Response · Authors · 2024-11-25
**General Response**

## **General Response (1/6)**

We sincerely appreciate the time, effort, and detailed comments from the Reviewers.

We first respond to the four common issues in the reviews and then respond separately to each Reviewer.\
&nbsp;
&nbsp;
&nbsp;
&nbsp;
### **G0. What is the novelty and contribution of this work?**

This paper focuses on **compressing the Small Language Models (SLMs)** [8,9] **deployed on low-end devices (i.e. Raspberry Pi) in edge applications.**
The edge applications pose two requirements to our compressing approach, which are not common in LLM applications:
- **Adaptivity**: the approach should dynamically adjust the model to the environmental changes (e.g. tokens registered or deregistered);
- **Low energy**: the computation and memory operations should consider the energy consumption (i.e. for longer battery life).

Centred on these two issues (detailed discussion is in Section 2 of the updated submission), our approach based on Tensor-Train Decomposition (TTD) is **specifically designed for compressing SLMs**:
1. Adaptivity:  TTD works on embedding vector level, which allows the application to update the vocabulary without operating on the whole compressed embedding matrix;
2. Low energy: As computation operations are "cheaper" than memory operations regarding energy consumption, we chose to "exchange" the memory with computation to save energy during the forwarding passes, with negligible extra latency.

As far as we know, none of the current LLM (at least for those designed for GPUs) compression work has these concerns, though these concerns are critical for low-end devices and edge applications.
TT format has an expressive and flexible form, which makes it easier for us to analyze and satisfy the two requirements.

Apart from common metrics like compression ratio and language task performance, we also give the estimated **energy costs** (with a similar approach in [18]) of our approach and SVD-based approach. At a quick glance, the following is the estimated inference energy costs with their best language task performance are

&nbsp;
&nbsp;

**Tab.G.0 Inference energy costs of an input text of 100 tokens.** (% percentage of uncompressed model energy costs, the lower, the better)
| Model               | OPT-125M | OPT-350M | DistilGPT2 | GPT-2  | GPT2-M | GPT2-L | CerebrasGPT-111M | CerebrasGPT-256M | CerebrasGPT-590M |
|---------------------|------------:|------------|------------:|--------|--------:|--------|--------------------:|--------------------|--------------------:|
| **SVD**  | 84.44%     | **43.35%**     | 70.28%     | 70.28% | 54.01% | 44.24% | 84.44%             | 81.80%             | 75.34%             |
| **Ours** | **59.21%**     | 52.73%     | **66.84%**     | **59.21%** | **52.74%** | **43.23%** | **61.51%**             | **51.71%**             | **50.45%**             |
| ||  | | |

The details of this part can be found in line 137-186, line 281-300, line 515-522 of the updated submission.

&nbsp;
&nbsp;
&nbsp;
&nbsp;
### **G1. The investigated LLMs are too small.**

We feel there might be some misunderstanding here, especially since our paper focuses on the sub-billion language models (as stated in MobileLLM [8]) running on **low-end devices**, and we had stated this in the abstract and introduction of our original submission.

We have therefore decided to change the title of our paper to avoid this ambiguity. The new title is **"TensorSLM: Sub-billion Parameter Language Model Compression for Low-end Devices based on Tensor-Train Decomposition"**. We have also rephrased our paper to further emphasise the focus (mainly Section 1,2,3 in the updated submission).

---

> ### Author Response · Authors · 2024-11-25
>
> ## **General Response (2/6)**
> &nbsp;
> &nbsp;
> ### **G2. Lack of the comparison of relevant works.**
> We did address our differences with the relevant works in line 42 - 51 in our original submission. Along with the referred works addressed by the Reviewers, the comparison is as following:
>
> **Tab.G.2 Study on LM compression or relevant low-rank factorization**
> | ||  | | | |  | ||  |
> |----------------------------------------------------------------|-----------------------:|-----------------------|----------------|---------------------:|--------------------|---------------:|------------|------------------------:|------------------------|
> | | **high-end device** | **low-end device** | **Training required?** | **Matrix** | **Tensor** | **Embedding layer** | **Linear layer** | **LLMs** | **SLMs** |
>  GroupReduce [12]            | $\surd$                     |                       | $\surd$              | $\surd$                   |                    | $\surd$             |            |                        | $\surd$                      |
> | [11]                | $\surd$                     |                      |    $\surd$            |                     | $\surd$                  | $\surd$             |            |                        | $\surd$                      |
> | [17]                | $\surd$                     |                      |    $\surd$            |                     | $\surd$                 |              | $\surd$          |       $\surd$                 |                       |
> | LightToken [7]                             | $\surd$                     |                       | $\surd$              | $\surd$                   |                    | $\surd$             |            |                        | $\surd$                      |
> | DSVD [10]                                   |                       | $\surd$                     |    $\surd$             | $\surd$                   |                    | $\surd$             |            |                        | $\surd$                      |
> | iRVQ[13]                               | -                     | -                     | $\surd$              | $\surd$                   |                    | -             | -          | -                      | -                      |
> | DCQ [14]                                 | $\surd$                     |                       | $\surd$              | $\surd$                   |                    | $\surd$             |            |                        | $\surd$                      |
> | ASVD[4]                              | $\surd$                     |                       |                | $\surd$                   |                    |               | $\surd$          | $\surd$                      |                        |
> | [3]     | $\surd$                     |                       | $\surd$              | $\surd$                   |                    |               | $\surd$          |                        | $\surd$                      |
> | ModeGPT[5]                          | $\surd$                     |                       |                | $\surd$                   |                    |               | $\surd$          | $\surd$                     |                        |
> | Monarch[15]                        | $\surd$                     |                       | $\surd$              | $\surd$                   |                    |               | $\surd$          |                        |                        |
> | [16]                               | $\surd$                     |                       | $\surd$              | $\surd$                   | $\surd$                 |               | $\surd$          |                        |                        |
> | MobileLLM[8]                     | $\surd$                     |                       | $\surd$              | -                   | -                  | -             | -          |                        | $\surd$                      |
> | **Ours**                                            |                       | $\surd$                     |                |                     | $\surd$                  | $\surd$             |            |    |$\surd$                      |                        |
> ||||
>
> It should be noticed that though SliceGPT [6] also works with matrices, it performs a kind of pruning that exploits the sparsity rather than low-rank. Thus, it is outside the scope of low-rank factorization and the relevant works.
>
> From Tab.G.2 we can observe that none of the relevant works has the same focus as ours. Though [10-12, 15-16] do not require fine-tuning the compressed model, they need to train a meta-model to get the compressed weights. If the input token distribution changes, the meta-models still require fine-tuning. Thus, for edge applications in this paper, the meta-learning based approaches [10-12, 15-16] are outside our scope of "training-free".

---

> ### Author Response · Authors · 2024-11-25
>
> ## **General Response (3/6)**
> &nbsp;
> &nbsp;
> ### **G3. Evaluation of more complex language tasks and LMs outside the GPT family is required.**
>
> We have extended our experiments with OPT-{125M, 350M, 1.3B} on zero-shot reasoning tasks with APIs in [1], since OPT performs well on the zero-shot reasoning tasks (as shown in Fig.1 of the updated submission). The experimental results are as follows. The **bold** numbers indicate the top-3 best performance cases. We also evaluated CerebrasGPT on these tasks, which is available in our updated submission.
>
> We can observe from Tab. G.3.1 - G.3.7 that our approaches have a higher chance of maintaining the language task performance (especially in the average scores).
>
> **Tab. G.3.1 OPT-125M**
> |                | Param (%) | ARC-c | ARC-e | BoolQ | HellaS.   | PIQA     | SIQA       | WinoG.   | Avg.  |
> |------------------|-----------|-------|-------|-------|-----------|----------|------------|----------|-------|
> | Original         | 100.00    | 23.38 | 57.11 | 57.74 | 41.53     | 71.71    | 34.49      | 59.35    | 49.33 |
> | SVD  (matrices)  | 0.13      | **21.5**  | 25.84 | 37.83 | 25.81     | 52.5     | 32.91      | 49.01    | 35.06 |
> |                  | 26.57     | 20.05 | 31.14 | 37.83 | 26.59     | 56.31    | **34.03**      | 50.59    | 36.65 |
> |                  | 53.01     | 18.17 | 34.26 | 37.83 | 26.96     | 57.56    | 33.78      | **52.88**    | 37.35 |
> |                  | 79.45     | 18.77 | **39.9**  | 45.63 | 27.38     | **59.9**     | **34.34**      | 50.83    | 39.54 |
> |                  | 92.67     | 18.77 | **43.14** | **47.37** | **28.51**     | **63**       | **34.14**      | **51.3**    | **40.89** |
> | Ours  (vectors)  | 2.47      | **21.33** | 26.39 | 37.83 | 25.63     | 52.94    | 33.98      | 50.59    | 35.53 |
> |                  | 29.17     | 20.22 | 28.66 | 39.14 | 26.17     | 53.54    | 33.83      | 49.88    | 35.92 |
> |                  | 50.78     | **21.25** | 29.55 | 40.15 | 26.19     | 54.9     | 33.37      | 50.12    | 36.50 |
> |                  | 71.88     | 19.37 | 35.31 | **47.09** | **27.78**     | 59.58    | 33.37      | 50.59    | **39.01** |
> |                  | 87.11     | 19.03 | **39.6**  | **59.51** | **28.41**     | **61.15**    | 33.78      | **51.14**    | **41.80** |
> ||||||
>
> \
> \
> \
> **Tab. G.3.2 OPT-350M**
>
> |                 | Param (%) | ARC-c | ARC-e | BoolQ | HellaS.   | PIQA     | SIQA       | WinoG.   | Avg.  |
> |------------------|-----------|-------|-------|-------|-----------|----------|------------|----------|-------|
> | Original         | 100.00    | 20.82 | 44.19 | 57.68 | 32.03     | 64.64    | 32.96      | 52.09    | 43.49 |
> | SVD  (matrices)  | 0.20      | **21.5**  | 25.25 | 37.83 | 25.67     | 51.36    | 32.32      | 49.17    | 34.87 |
> |                  | 19.93     | **20.82** | 25.93 | 38.53 | 25.94     | 53.92    | 32.11      | 51.07    | 35.62 |
> |                  | 39.66     | 20.22 | 25.8  | 38.62 | 26.22     | 53.26    | 32.16      | 50.59    | 35.41 |
> |                  | 59.39     | 19.2  | 25.55 | 38.5  | 26.54     | 53.97    | 33.03      | **51.46**    | 35.61 |
> |                  | 79.12     | 19.2  | **27.53** | 37.83 | **27.16**     | **55.93**    | 32.62      | 49.33    | 35.80 |
> |                  | 98.85     | 20.73 | **41.37** | 37.89 | **30.47**     | **62.95**    | 32.73      | 49.25    | **39.48** |
> | Ours  (vectors)  | 3.52      | **21.08** | 24.92 | 45.47 | 25.66     | 53.1     | **33.7**       | 51.14    | 36.58 |
> |                  | 18.75     | 20.39 | 26.01 | **62.17** | 25.9      | 53.16    | 32.16      | 48.7     | 38.50 |
> |                  | 28.13     | 20.05 | 24.87 | **62.17** | 26.08     | 53.54    | **33.14**      | 49.33    | 38.60 |
> |                  | 42.19     | 20.05 | 25.25 | 48.44 | 26.18     | 53.7     | 32.32      | 49.09    | 36.58 |
> |                  | 70.31     | 20.48 | 25.42 | **62.17** | 26        | 53.16    | 32.27      | **51.62**    | **38.87** |
> |                  | 94.53     | **21.42** | **36.15** | 45.9  | **29.59**     | **61.92**    | **33.14**      | **52.33**    | **40.21** |
> ||||||

---

> ### Author Response · Authors · 2024-11-25
>
> ## **General Response (4/6)**
> **Tab. G.3.3 OPT-1.3B**
>
> |                | Param (\%) | ARC-c | ARC-e | BoolQ | HellaS. | PIQA  | SIQA  | WinoG. | Avg.  |
> |-----------------|------------|-------|-------|-------|---------|-------|-------|--------|-------|
> | Original        | 100.00     | 23.38 | 57.11 | 57.74 | 41.53   | 71.71 | 34.49 | 59.35  | 49.33 |
> | SVD  (matrices) | 0.05       | 21.93 | 26.43 | 38.75 | 25.66   | 53.75 | 33.32 | 49.8   | 35.66 |
> |                 | 25.46      | 20.31 | 34.85 | 40.89 | 26.46   | 57.07 | 34.29 | 50.91  | 37.83 |
> |                 | 50.87      | 20.56 | 44.87 | 57.34 | 28.04   | 63.28 | **35.52** | 51.3   | 42.99 |
> |                 | 76.28      | 22.01 | 50.13 | **61.8**  | 30.69   | 66.76 | 34.65 | **56.51**  | 46.08 |
> |                 | 96.61      | **23.55** | **53.96** | **60.28** | **36.35**   | **69.59** | 34.7  | 57.77  | **48.03** |
> | Ours  (vectors) | 1.07       | 21.33 | 25.38 | 42.69 | 25.39   | 53.32 | 33.98 | 50.28  | 36.05 |
> |                 | 24.22      | 21.16 | 25.93 | **60.43** | 25.88   | 54.9  | 34.54 | 50.36  | 39.03 |
> |                 | 49.41      | 21.08 | 26.35 | 54.04 | 25.91   | 53.81 | 34.54 | 48.93  | 37.81 |
> |                 | 70.70      | **25.26** | **52.86** | 57.98 | **38.78**   | **69.48** | **35.31** | **58.48**  | **48.31** |
> |                 | 94.73      | **23.38** | **55.22** | 51.68 | **40.43**   | **71**    | **35.31** | **59.43**  | **48.06** |
> ||||||
>
> \
> \
> \
> **Tab. G.3.4 CerebrasGPT-111M**
>
> |               | Param (%) | ARC-c | ARC-e | BoolQ | HellaS.   | PIQA     | SIQA       | WinoG.   | Avg.  |
> |------------------|-----------|-------|-------|-------|-----------|----------|------------|----------|-------|
> | Original         | 100.00    | 16.64 | 37.88 | 62.14 | 26.76     | 59.41    | 33.88      | 49.01    | 40.82 |
> | SVD  (matrices)  | 0.13      | **20.9**  | 26.52 | 37.86 | 25.46     | 52.45    | 33.57      | 48.93    | 35.10 |
> |                  | 26.57     | 17.49 | 31.44 | 37.77 | 26.53     | 56.2     | 33.57      | **50.99**    | 36.28 |
> |                  | 53.01     | 17.06 | **35.27** | 44.16 | 26.57     | 56.75    | 34.08      | **50.28**    | 37.74 |
> |                  | 79.45     | 16.55 | **37.08** | 59.88 | **26.76**     | **58.81**    | 33.88      | 49.72    | **40.38** |
> |                  | 92.67     | 15.44 | **37.92** | **61.77** | **26.84**     | **59.19**    | 33.62      | 49.17    | **40.56** |
> | Ours  (vectors)  | 2.47      | 19.97 | 28.24 | **61.93** | 26.09     | 54.13    | **34.54**      | **50.36**    | 39.32 |
> |                  | 29.17     | **20.48** | 29.84 | 59.85 | 26.26     | 55.66    | **34.7**       | 50.04    | 39.55 |
> |                  | 50.78     | 19.8  | 31.48 | 49.11 | **26.78**     | 57.51    | 33.42      | 49.09    | 38.17 |
> |                  | 71.88     | 17.92 | 34.51 | 58.32 | 26.74     | **58.05**    | **34.54**      | **50.28**    | **40.05** |
> |                  | 87.11     | **20.99** | 24.07 | **61.04** | 25.66     | 52.67    | 33.98      | 49.49    | 38.27 |
> ||||
>
> \
> \
> \
> **Tab. G.3.5 CerebrasGPT-256M**
>
> |      256           | Param (%) | ARC-c | ARC-e | BoolQ | HellaS.   | PIQA     | SIQA       | WinoG.   | Avg.  |
> |-----------------|-----------|-------|-------|-------|-----------|----------|------------|----------|-------|
> | Original        | 100.00    | 16.89 | 40.95 | 61.5  | 27.44     | 61.37    | 34.24      | 51.3     | 40.82 |
> | SVD (matrices)  | 0.09      | **21.16** | 26.73 | 37.83 | 25.74     | 52.29    | 32.8       | 51.22    | 35.40 |
> |                 | 28.26     | 17.75 | 33    | 38.2  | 26.48     | 58.05    | 33.98      | 50.43    | 36.84 |
> |                 | 47.05     | 17.15 | 35.61 | 39.97 | 26.83     | 59.03    | **34.19**      | **51.78**    | 37.79 |
> |                 | 75.22     | 18.09 | **39.44** | **61.01** | **27.4**      | **60.83**    | **34.03**      | **51.62**    | **41.77** |
> |                 | 94.00     | 18.17 | **40.74** | **59.94** | **27.4**      | **61.04**    | 33.67      | 50.91    | **41.70** |
> | Ours (vectors)  | 2.67      | **20.22** | 24.62 | 37.74 | 25.5      | 54.57    | **34.19**      | 50.75    | 35.37 |
> |                 | 38.60     | **20.9**  | 27.57 | 37.83 | 25.8      | 53.48    | 33.47      | 49.96    | 35.57 |
> |                 | 50.37     | 18.94 | 31.52 | 52.35 | 26.76     | 56.86    | 33.93      | 50.67    | 38.72 |
> |                 | 61.40     | 20.05 | 35.02 | 59.91 | 27.3      | 57.94    | 33.88      | 50.59    | 40.67 |
> |                 | 98.90     | 19.28 | **40.74** | **61.5**  | **27.39**     | **61.43**    | 33.98      | **52.01**    | **42.33** |
> |||

---

> > ### Author Response · Authors · 2024-11-25
> >
> > ## **General Response (5/6)**
> > **Tab. G.3.6 CerebrasGPT-590M**
> >
> > |         | Param (%) | ARC-c | ARC-e | BoolQ | HellaS.   | PIQA     | SIQA       | WinoG.   | Avg.  |
> > |-----------------|-----------|-------|-------|-------|-----------|----------|------------|----------|-------|
> > | Original        | 100.00    | 19.03 | 46.42 | 59.17 | 29.12     | 62.73    | 35.31      | 49.8     | 43.08 |
> > | SVD (matrices)  | 0.07      | **21.33** | 27.1  | 37.92 | 25.79     | 52.45    | 34.03      | 47.99    | 35.23 |
> > |                 | 26.91     | 18.09 | 35.98 | 37.83 | 26.82     | 58       | 34.14      | 50.59    | 37.35 |
> > |                 | 47.03     | 17.24 | 39.31 | 37.83 | 27.56     | 59.74    | 34.75      | **50.83**    | 38.18 |
> > |                 | 73.87     | 18.86 | **44.11** | 49.45 | **28.42**     | **61.64**    | **34.8**       | **50.75**    | 41.15 |
> > |                 | 94.00     | 19.8  | **46.17** | 52.72 | **28.97**     | **62.19**    | **35.62**      | 49.88    | **42.19** |
> > | Ours (vectors)  | 1.37      | **23.38** | 24.87 | **56.61** | 25.59     | 52.67    | 33.73      | **52.17**    | 38.43 |
> > |                 | 19.21     | 19.97 | 26.81 | 49.82 | 25.66     | 52.72    | 34.24      | 49.17    | 36.91 |
> > |                 | 46.88     | 19.54 | 35.9  | 40.89 | 27.04     | 57.83    | 34.54      | 49.72    | 37.92 |
> > |                 | 66.41     | 20.31 | 38.09 | **58.87** | 28.21     | 60.28    | 34.29      | 49.88    | **41.42** |
> > |                 | 94.34     | **22.1**  | **44.7**  | **56.42** | **29.02**     | **61.64**    | **35.52**      | 49.49    | **42.70** |
> > ||||
> >
> >
> > \
> > \
> > \
> > **Tab. G.3.7 CerebrasGPT-1.3B**
> > |              | Param (%) | ARC-c | ARC-e | BoolQ | HellaS.   | PIQA     | SIQA       | WinoG.   | Avg.  |
> > |------------------|-----------|-------|-------|-------|-----------|----------|------------|----------|-------|
> > | Original         | 100.00    | 22.35 | 50.88 | 59.33 | 32.55     | 66.49    | 34.44      | 51.93    | 45.42 |
> > | SVD  (matrices)  | 0.05      | 21.33 | 26.73 | 40.06 | 25.65     | 52.39    | 33.32      | 48.86    | 35.48 |
> > |                  | 25.46     | 18    | 38.59 | 37.83 | 27.38     | 59.3     | **34.95**      | **51.85**    | 38.27 |
> > |                  | 50.87     | 19.71 | 45.08 | 50.49 | 29.13     | 62.35    | 34.29      | **52.25**    | 41.90 |
> > |                  | 76.28     | 19.97 | **49.03** | **55.6**  | 30.7      | 64.85    | 34.49      | 49.96    | 43.51 |
> > |                  | 96.61     | 20.99 | **50.51** | 53.82 | **32.06**     | **65.61**    | **34.54**      | 50.43    | **43.99** |
> > | Ours  (vectors)  | 1.07      | **22.53** | 27.95 | 40.03 | 25.86     | 54.35    | 34.24      | 50.75    | 36.53 |
> > |                  | 24.22     | 21.42 | 27.36 | 39.54 | 25.78     | 53.37    | 33.78      | 50.59    | 35.98 |
> > |                  | 57.03     | **22.61** | 39.56 | 53.52 | 30.44     | 63.06    | 33.62      | 47.75    | 41.51 |
> > |                  | 70.70     | **22.61** | 43.6  | **61.35** | **31.4**      | **65.07**    | **34.95**      | 50.04    | **44.15** |
> > |                  | 94.73     | 22.35 | **46.97** | **56.36** | **32.21**     | **65.61**    | 33.98      | **51.78**    | **44.18** |
> > ||||

---

> ### Author Response · Authors · 2024-11-25
>
> ## **General Response (6/6)**
> &nbsp;
> &nbsp;
> ### **Reference**
>
> [1] "Tensor-train decomposition". SIAM Journal on Scientific Computing, 33(5):2295–2317, 2011. doi: 10.1137/090752286.
>
> [2] "A framework for few-shot language model evaluation." Version v0. 0.1. Sept 10 (2021): 8-9.
>
> [3] "Language model compression with weighted low-rank factorization." ICLR 2022.
>
> [4] "Asvd: Activation-aware singular value decomposition for compressing large language models." arXiv preprint arXiv:2312.05821 (2023).
>
> [5] "MoDeGPT: Modular Decomposition for Large Language Model Compression." arXiv preprint arXiv:2408.09632 (2024).
>
> [6] "Slicegpt: Compress large language models by deleting rows and columns." arXiv preprint arXiv:2401.15024 (2024).
>
> [7] "LightToken: A task and model-agnostic lightweight token embedding framework for pre-trained language models." Proceedings of the 29th ACM SIGKDD Conference on Knowledge Discovery and Data Mining. 2023.
>
> [8] "MobileLLM: Optimizing Sub-billion Parameter Language Models
> for On-Device Use Cases", ICML 2024
>
> [9] "Small Language Models: Survey, Measurements, and Insights." arXiv preprint arXiv:2409.15790 (2024).
>
> [10] "Direction is what you need: improving word embedding compression in large language models." arXiv preprint arXiv:2106.08181 (2021).
>
> [11] "Tensorized embedding layers for efficient model compression." arXiv preprint arXiv:1901.10787 (2019).
>
> [12] "Groupreduce: Block-wise low-rank approximation for neural language model shrinking." Advances in Neural Information Processing Systems 31 (2018).
>
> [13] "Improved Residual Vector Quantization for High-dimensional Approximate Nearest Neighbor Search." arXiv preprint arXiv:1509.05195 (2015).
>
> [14] "Learning k-way d-dimensional discrete codes for compact embedding representations." International Conference on Machine Learning. PMLR, 2018.
>
> [15] "Monarch: Expressive structured matrices for efficient and accurate training." International Conference on Machine Learning. PMLR, 2022.
>
> [16] "Compute Better Spent: Replacing Dense Layers with Structured Matrices." arXiv preprint arXiv:2406.06248 (2024).
>
> [17] "Efficient gpt model pre-training using tensor train matrix representation." arXiv preprint arXiv:2306.02697 (2023).
>
> [18] "Addition is all you need for energy-efficient language models." arXiv preprint arXiv:2410.00907 (2024).

---

### Meta-Review · Area_Chair_qc5E · 2024-12-23

**Metareview:**

This paper proposes TensorGPT, a method for compressing small language models (SLMs) using tensor-train decomposition (TTD) of the token embedding layer. The authors claim this approach is training-free and suitable for deploying SLMs on low-end devices. The method is evaluated on GPT-2, OPT, and CerebrasGPT models with up to 1.3B parameters.

The paper addresses the important issue of SLM compression for edge devices and provides experiments on low-end hardware like Raspberry Pi. The proposed training-free compression method and consideration of energy efficiency are relevant for resource-constrained scenarios.

However, the paper's novelty is limited, as tensor-train decomposition has been used before in language model compression. The lack of comparisons to state-of-the-art baselines and the focus on relatively small models limit the generalizability and impact of the work. Additionally, the initial claims about being first to compress LLMs with low-rank factorization were overstated.

The primary reasons for rejection are the paper's failure to demonstrate significant technical novelty or empirical contributions beyond applying existing techniques to a specific use case. While the work addresses an important problem, it falls short of the level of innovation and impact expected for publication at ICLR.

**Additional Comments On Reviewer Discussion:**

During the discussion, reviewers raised concerns about limited novelty, lack of comparisons to state-of-the-art baselines, focus on small models, and overstatement of claims. The authors responded by clarifying their focus on SLMs for low-end devices, adding comparisons to SVD and SliceGPT baselines, expanding experiments to include OPT models and more reasoning tasks, and emphasizing adaptivity and low-energy requirements.

Although these responses addressed some concerns, they did not fully alleviate the core issues. The expanded experiments and clarifications, while noted, did not significantly change the overall contribution. The focus on sub-1B parameter models still limits the broader applicability of the method, and the added baselines did not include some of the most recent and competitive approaches in the field.

In the final decision, the limited technical novelty and lack of substantial advances over existing compression techniques for SLMs were the primary factors for rejection. Despite addressing an important problem, the work does not meet the innovation and impact standards expected for ICLR publication.

---

### Decision · Program_Chairs · 2025-01-22

Reject